# ANALOGGENIE: A GENERATIVE ENGINE FOR AUTOMATIC DISCOVERY OF ANALOG CIRCUIT TOPOLOGIES

**Jian Gao**[1]**, Weidong Cao**[2]**, Junyi Yang**[1]**, Xuan Zhang**[1]
[1] Northeastern University, [2] The George Washington University
{gao.jian3,yang.juny,xuan.zhang}@northeastern.edu
{weidong.cao}@gwu.edu

## ABSTRACT

The massive and large-scale design of foundational semiconductor integrated circuits (ICs) is crucial to sustaining the advancement of many emerging and future technologies, such as generative AI, 5G/6G, and quantum computing. Excitingly, recent studies have shown the great capabilities of foundational models in expediting the design of digital ICs. Yet, applying generative AI techniques to accelerate the design of analog ICs remains a significant challenge due to critical domain-specific issues, such as the lack of a comprehensive dataset and effective representation methods for analog circuits. This paper proposes, **AnalogGenie**, a **Gen**erativ**e** **e**ngine for automatic design/discovery of **Analog** circuit topologies–the most challenging and creative task in the conventional manual design flow of analog ICs. AnalogGenie addresses two key gaps in the field: building a foundational comprehensive dataset of analog circuit topology and developing a scalable sequence-based graph representation universal to analog circuits. Experimental results show the remarkable generation performance of AnalogGenie in broadening the variety of analog ICs, increasing the number of devices within a single design, and discovering unseen circuit topologies far beyond any prior arts. Our work paves the way to transform the long-standing time-consuming manual design flow of analog ICs to an automatic and massive manner powered by generative AI. Our source code is available at https://github.com/xz-group/AnalogGenie.

## 1 INTRODUCTION

Semiconductor integrated circuits (ICs) are the foundational hardware cornerstone to advance many emerging technologies such as generative AI, 5G/6G, and quantum computing. The demand for and the scale of ICs are soaring to unprecedented levels with the ever-increasing information and computing workloads (e.g., training foundation models with billions of parameters) (Achiam et al., 2023). Thus, accelerating the design of advanced ICs is a key to sustaining the development of future technologies. Excitingly, recent breakthroughs in generative AI have presented transformative opportunities to expedite the conventional design flows of ICs. Domain-specific large language models (LLMs) have been developed to free human designers by automatically generating and correcting Hardware Description Languages (HDL) (Zhong et al., 2023; Blocklove et al., 2023; Chang et al., 2023; Thakur et al., 2024; 2023; Fu et al., 2023; Liu et al., 2023b; Wu et al., 2024; Liu et al., 2023a), which can be seamlessly used to synthesize digital ICs with desired functionalities. As an example, NVIDIA's ChipNeMo (Liu et al., 2023a), a powerful domain-adapted LLM, can rapidly generate valuable digital designs with just a few prompts. Yet, applying generative AI to speed up the design of analog ICs–essential in ubiquitous electronic systems to bridge the interfaces between the physical world and cyberspace, ranging from enhancing performance in computing systems (e.g., high-speed memory interfaces and I/O links) to providing critical functionalities in communication and sensing systems (e.g., 5G/6G and quantum computing)–remains significantly understudied.

The fundamental challenge arises from the intricate design complexities of analog ICs. Unlike digital ICs that can be universally and hierarchically abstracted into Boolean logic representations and easily described with high-level hardware description languages (e.g., Verilog and VHDL) or programming languages (e.g., C), analog ICs remain intractable to such abstraction due to their

lack of systematic hierarchical representation and the heuristic and knowledge-intensive nature of their design process (Gielen & Rutenbar, 2000). This makes it extremely hard to automate the design of analog ICs by developing programming languages similar to those used for digital ICs. As such, domain experts have followed a longstanding manual flow to design analog ICs. This process involves a number of time-consuming stages, such as selecting/creating an existing (new) circuit topology (i.e., defining the connections between devices), optimizing device parameters based on the topology to achieve desired performance, and designing the physical layout of the optimized circuit for manufacturing. Importantly, the topology generation stage is the foundation and most creative part of the analog IC design process, posing a formidable and perennial challenge to design automation. Addressing it is the key to accelerating the development of analog ICs.

There have been several studies in tackling this problem with generative AI techniques. The early pioneering work, CktGNN (Dong et al., 2023), formulates the topology design as a graph generation task, as circuit topologies of analog ICs can be naturally represented as graph structures. It uses a graph variational autoencoder (VAE) to generate various circuit topologies for a specific type of analog ICs, i.e., operational-amplifiers (Op-Amps). More recently, foundational models have also been explored for designing analog circuit topologies. LaMAGIC (Chang et al., 2024), a fine-tuned masked language model (MLM), has been proposed to generate analog circuits with a fixed number of graph nodes. It shows a high success rate in designing a specific type of analog ICs, i.e., power converters (with fewer than 4 devices). AnalogCoder (Lai et al., 2024), another LLM-based work, uses domain-specific prompt engineering to generate analog circuits from well-established LLM models (e.g., GPT-4). Instead of directly generating circuit topologies, it generates PySpice codes that can be converted to a SPICE (Simulation Program with Integrated Circuit Emphasis) netlist–a textual high-level description of device connections used for circuit simulation. AnalogCoder can generate a range of conventional analog circuits that often have a limited number of devices on the order of ten. These methods have demonstrated the potential of applying generative AI to analog IC design. Yet, a vast untapped frontier remains.

This work proposes, **AnalogGenie**, a **Gen**erati**ve e**ngine (model) for automatic discovery of **analog** circuit topologies. In contrast to previous methods (Dong et al., 2023; Chang et al., 2024; Lai et al., 2024) that are limited to a smaller scale of generation (e.g., generating a single type of analog ICs, small-size analog ICs, or conventional analog ICs), **AnalogGenie addresses the problem of scalable and general design**. It can significantly broaden the variety of analog ICs, increase the number of devices within a single design, and discover unseen circuit topologies. A major obstacle to advancing generative models for scalable analog circuit design automation is the lack of a comprehensive dataset of analog circuit topologies. We bridge this gap by building a extensive dataset that consists of more than 3000 distinct analog circuit topologies with diverse functionalities (e.g., Op-Amps, Low Dropout Regulator (LDO), Bandgap reference, Comparator, Phase-Locked Loop (PLL), Low Noise Amplifier (LNA), Power Amplifiers (PA), Mixer, Voltage-Controlled Oscillator (VCO), etc) from public resources (Razavi, 2000; Razavi & Behzad, 2012; Johns & Martin, 2008; Gray et al., 2009; Allen & Holberg, 2011; Camenzind, 2005). In addition, we apply data augmentation techniques to expand these circuit topologies by over $70\times$. To the best of our knowledge, this is the largest circuit dataset that effectively incorporates and enhances existing real-world analog circuit topologies to the greatest extent. This enables AnalogGenie to effectively learn various analog topologies and significantly enhance its generation capabilities, surpassing all previous methods (Dong et al., 2023; Chang et al., 2024; Lai et al., 2024).

Nonetheless, another key barrier to advancing the scalable design of analog circuits is short of a scalable and unambiguous representation of circuit topologies. AnalogCoder (Lai et al., 2024) relies on high-level text representations that use multiple tokens to describe a single connection between devices, making the generation prone to errors. CktGNN (Dong et al., 2023) and LaMAGIC (Chang et al., 2024) use graph-based representations with a fixed number of nodes, where each node represents a circuit device or subgraph. This ignores the critical low-level details essential in analog circuit design, leading to ambiguous and unscalable circuit generation. We propose a scalable sequence-style data structure that captures fundamental analog circuit design details while efficiently describing large circuit graphs. Specifically, we represent each circuit topology as an undirected graph where each node is a device pin (Figure 3). We then sequentialize it into an Eulerian circuit–a trail that visits every edge exactly once and starts and ends at the same node. This unique representation allows AnalogGenie to generate circuit topologies in a scalable, flexible, and efficient manner. These developed dataset and techniques can thus enable AnalogGenie with excep-

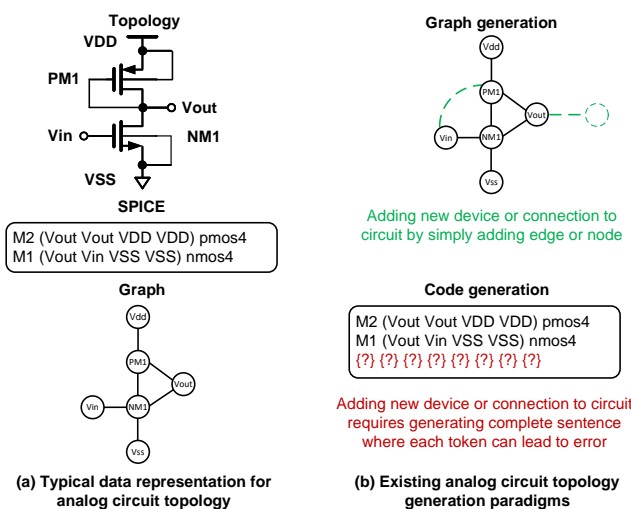

Figure 1: Current states of analog circuit topology generation. (a) Typical data representation for analog circuit topology. (b) Existing analog circuit topology generation paradigms. Graph provides a clear one-to-one mapping between the graph generation process and the circuit design process. PySpice code is a high-level representation, making its generation process more prone to error.

tional capabilities to produce diverse, large, and unseen analog circuit topologies. The advancement holds both profound engineering and scientific significance, demonstrating that generative AI can not only meet human expertise but also unlock the possibilities beyond human capability.

The key contributions in this paper are: (1) We propose a generative engine, AnalogGenie, built on a GPT model to generate diverse analog circuits by predicting the next device pin to connect in the circuit; (2) We introduce a sequence-based, pin-level graph representation that efficiently and expressively captures large analog circuit topologies; (3) We develop a comprehensive dataset of analog circuit topologies to advance research in analog electronic design automation using generative AI and introduce an augmentation scheme to enhance data diversity. (4) Experiment results show that AnalogGenie is capable of automatically generating far more, large-scale, valid, unseen, and high-performance topologies compared to existing graph generation and foundation model work.

## 2 PRELIMINARIES AND RELATED WORKS

### 2.1 DESIGN PROCESSES OF ANALOG CIRCUITS

The design process of analog circuits begins with creating the circuit topology, which involves determining the device types (i.e., NMOS/PMOS transistor, capacitor, resistor, inductor, etc.) and the number of devices, and defining how they are interconnected. Following this, designers perform device sizing, i.e., optimizing the physical dimensions of devices to achieve desired performance. Finally, the physical layout (i.e., mask design) is developed to prepare for manufacturing. Note that a physical design is the representation of an IC in terms of planar geometric shapes corresponding to the different stacked physical layers (e.g., metal, oxide, or semiconductor) during the fabrication process. Of all these stages, topology design demands the most creative effort, as it needs to be conceptualized from scratch by human designers. While significant progress has been made in automating device sizing (Wang et al., 2020; Cao et al., 2022; Gao et al., 2023; Cao et al., 2024) and layout design (Kunal et al., 2019; Xu et al., 2019), the topology generation remains a challenging problem due to its abstract and complex nature. This work aims to address this thorny issue.

### 2.2 GENERATIVE AI FOR ANALOG CIRCUIT TOPOLOGY GENERATION

An analog circuit topology can be naturally represented as a graph structure, providing a clear one-to-one mapping between graph generation and topology design (Figure 1(a)). For instance, adding

a node to a graph corresponds directly to adding a new device to a circuit topology, while adding an edge between nodes represents a new connection between devices. This intuitive representation has led most existing topology generation methods to focus on graph generation (Figure 1(b)), such as CktGNN (Dong et al., 2023) and LaMagic (Chang et al., 2024). Yet, these approaches are often limited to generating only a single type of circuit topology (e.g., Op-Amps or power converters). This is because they rely on one-shot generation by predicting the adjacency matrix directly and pre-define the number of nodes for generation, thereby suffering from low scalability (Zhu et al., 2022). In contrast, our work employs sequential graph generation (Section 3), offering far greater flexibility and more adaptability to various types of circuit designs.

An analog circuit can also be compiled into a SPICE (Simulation Program with Integrated Circuit Emphasis) netlist (Figure 1(a)). A SPICE netlist is a text-based high-level description of the connection between devices (i.e., nets) in a circuit topology, which is used in the process of circuit performance simulation. Leveraging the powerful code and text generation capabilities of LLMs, recent work, AnalogCoder (Lai et al., 2024), applies domain-specific prompt engineering to existing LLMs to generate Python-style SPICE (i.e., PySpice) netlists for analog circuits. However, the availability of publicly accessible SPICE netlist data remains significantly limited compared to the wealth of publicly available analog circuit topologies. This is because analog circuit topologies are human-readable illustrations commonly found in textbooks and scientific publications, whereas netlists are software-oriented representations often used together with confidential semiconductor technologies to extract circuit performance by simulation tools. Another key challenge faced by code generation approaches is their reliance on high-level text-based circuit topology representations. Specifically, to add a new device or connection, autoregressive models must predict multiple tokens to generate a complete line of code (Figure 1(b)), making them more prone to errors compared to graph-based methods, which require only a single action per step. AnalogCoder (Lai et al., 2024) shows that even advanced models (e.g., GPT-4) struggle to correctly generate simple circuits with fewer than 10 devices. Thus, our work focuses on graph generation to achieve a more robust and scalable generation.

## 2.3 Open-Source Analog Circuit Datasets

The lack of a comprehensive analog circuit dataset fundamentally hinders the development of generative AI-based methods to automate the design of analog ICs. While some circuit datasets exist in the field, such as those provided by Align (Kunal et al., 2019), CktGNN (Dong et al., 2023), and AMSNet (Tao et al., 2024), they are often limited to specific types of analog circuits (i.e., Op-Amp) without any label (e.g., circuit performance). In addition, most of their topologies are synthesized by permutating pre-defined template, resulting in non-unique designs. To address this fundamental gap, we have created a thorough dataset by collecting 3350 distinct analog circuit topologies with diverse functionalities (e.g., LDO, Bandgap reference, Comparator, PLL, LNA, PA, Mixer, VCO, etc) from public resources (Razavi, 2000; Razavi & Behzad, 2012; Johns & Martin, 2008; Gray et al., 2009; Allen & Holberg, 2011; Camenzind, 2005). To ensure accurate connections, each schematic is manually drawn in an industry-standard circuit design tool for performance simulation. We also labeled each circuit with its performance metrics.

## 3 Approach

AnalogGenie is a domain-specific GPT model designed to generate various analog circuit topologies with greatly improved scalability. To achieve this, we first introduce an expressiveness-enhanced graph representation that models each device pin as an individual node, ensuring that every connection and interaction between circuit devices is explicitly represented. Next, we develop a sequence-style data structure to effectively handle large-scale analog circuits that can be typically modeled as large and sparse graphs. To further enhance the generation quality of AnalogGenie, we propose a data augmentation technique to address both data scarcity and the permutation invariance issue inherent in sequence data. Building upon these innovations, we customize a tokenizer to pre-train AnalogGenie and perform finetuning afterward, enabling AnalogGenie to generate specific type of high-performance circuits.

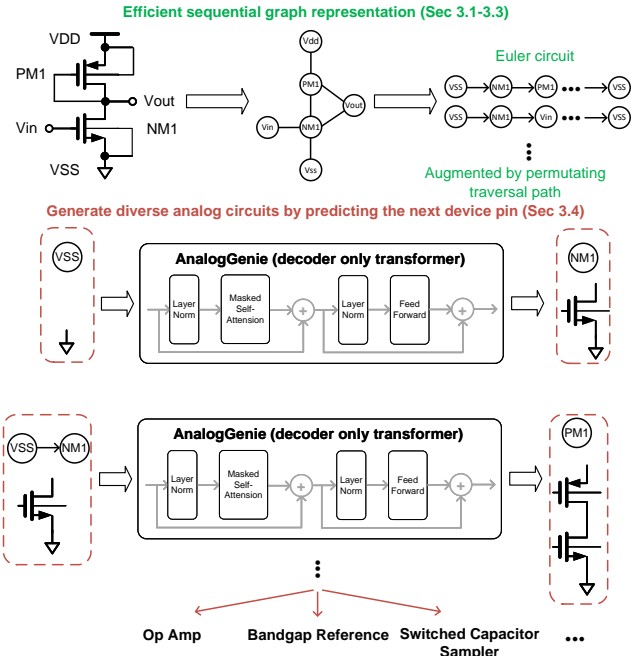

Figure 2: Overview of AnalogGenie. AnalogGenie represents each topology as a sequence and generates all sorts of analog circuit topology from scratch by predicting the device pin to connect.

## 3.1 EXPRESSIVENESS-ENHANCED GRAPH REPRESENTATION FOR TOPOLOGY MODELING

Prior works (Dong et al., 2023; Lu et al., 2023) rely on high-level graph representations to generate circuit topologies, where each node represents a device or a subgraph. This method omits essential low-level device details, leading to the issue of ambiguous generation, i.e., a single generated graph can be interpreted as multiple unique topologies. To understand this, consider that an NMOS transistor (NM) has four device pins–drain (D), gate (G), source (S), and body (B). When an entire device is abstracted into a single node, it becomes challenging to interpret to which pin an edge connects (Figure 3(a)). Therefore, to ensure a unique one-on-one mapping between the graph and the circuit topology–where every circuit connection is explicitly represented, an analog circuit has to be represented at the pin level (Figure 3(b)). Furthermore, previous methods restricted the graph representation of analog circuit topologies to directed acyclic graphs (DAGs), greatly limiting the types of circuit topologies that can be learned and generated. In this work, we adopt a more expressive and flexible representation of analog circuit topologies. Specifically, we represent the topology of an analog circuit as a finite connected undirected graph $\mathcal{G} = (V, E)$, where $V = \{1, 2, \ldots, n\}$ is the node set representing each device pin with $|V| = n$ and $E \in V \times V$ is the edge set. For each node $i$ in a graph $\mathcal{G}$, we let $\mathcal{N}(v) = \{u \in V \mid (u, v) \in E\}$ denote the set of neighboring nodes of $v$.

## 3.2 SEQUENTIAL GRAPH REPRESENTATION OF SCALABLE ANALOG CIRCUIT TOPOLOGIES

Previous methods (Dong et al., 2023; Lu et al., 2023) use adjacency matrices to represent circuit graphs. Yet, an adjacency matrix requires $O\left(n^2\right)$ space to store $n$ nodes, regardless of the number of edges, which is inefficient for sparse graphs. Analog circuit topologies are typically sparse because most devices are connected only to their immediate neighbors. As a result, the number of edges $e$ is far smaller than $n^2$, leaving the adjacency matrix filled with zeros and wasting significant space on non-existent edges. For example, the graph in Figure 2 has six nodes and six undirected edges or 12 directed edges. An adjacent matrix will need 6×6 matrices to represent them, wasting 24 elements to store nothing. In contrast, our work represents the graph as an Eulerian circuit that stores only existing edges, making it much more efficient than adjacency matrices, particularly for handling large analog circuit topologies. More examples of the advantages of using the Eulerian circuit to represent large sparse graphs can be found in Appendix A.3.

**Definition 3.2.1:** *Eulerian circuit is a graph trail that visits every edge exactly once and starts and ends at the same node.*

**Theorem 3.2.1:** *Let $\mathcal{G} = (V, E)$ be a finite connected undirected graph. Construct a directed graph $D = (V, A)$ by replacing each undirected edge $\{u, v\} \in E$ with two directed arcs $(u, v)$ and $(v, u)$ in A. Then, the directed graph D contains at least one Eulerian circuit starting from any node.*

*Proof.* Since directed graph $D = (V, A)$ is derived from a finite connected undirected graph $\mathcal{G} = (V, E)$, all of $D$'s node will have even degree and directed graph $D$ is connected. According to Euler theorem (Biggs et al., 1986), if a graph is connected and every node has even degree, then it has at least one Eulerian circuit. The Eulerian circuits can start at any vertex. Thus, the directed graph $D$ contains at least one Eulerian circuit starting from any vertex.

Theorem 3.2.1 clearly shows that all analog topologies have at least one Eulerian circuit once the original finite connected undirected graph $\mathcal{G}$ is converted into a finite connected directed graph $D$. Thus, we know that our Eulerian circuit can represent any analog circuits as long as they can be represented as finite connected undirected graph (i.e., Eulerian circuit traverse each directed edge exactly once when we convert finite connected undirected graph to finite connected directed graph).

### 3.3 Data Augmentation to Prevent Over-fitting and Reduce Inductive Bias

Conventional graph-based generation circuit methods (Dong et al., 2023; Chang et al., 2024) rely on synthetic circuit data to mitigate model overfitting by permutating circuit connections under a fixed number of nodes. Models trained on such data fail to capture the full complexity and nuances of real-world circuits, limiting their ability to generate only a single type of analog circuits. In addition to the overfitting, another critical issue that significantly impacts the model's learning ability is the inherent bias in data representation. Permutation invariance is a fundamental inductive bias of graph-structured data. For a graph with $n$ nodes, there are up to $n!$ different adjacency matrices that are equivalent representations of the same graph. A well-designed circuit generative model should assign the same probability to each of these equivalent representations.

To address these limitations, AnalogGenie learns from real-world circuits with diverse circuit types and their augmented representations. Specifically, AnalogGenie learns diverse representations from each analog circuit topology by generating multiple unique Eulerian circuits that represent the same topology. This allows us to generate $70\times$ more data. Eulerian circuits ensure each sequence traverses each directed edge exactly once. To enforce graph-level permutation invariance, AnalogGenie adopt the approach used by GraphRNN (You et al., 2018), employing a breadth-first-search (BFS) algorithm to assign node orders within each device type, ensuring that each circuit topology has a unique graph representation. To further minimize permutation invariance at the sequence level, AnalogGenie manually defines "VSS" (the ground node universal to all analog circuits) as the starting node for all sequences. These techniques let AnalogGenie to learn a robust and generalizable representation.

### 3.4 Customizing Tokenizer to Pre-Train a Domain-Specific GPT Model

Built on the above foundations, we customize a tokenizer to encode and decode our sequence that represents the circuit topology to train AnalogGenie. Table 2 in Appendix A.2 shows an example of a look-up table used for tokenization, where each token represent a device pin. The device type and the maximum number of devices in each device type are determined through a data-driven method by scanning the devices types and number in the training data. To allow AnalogGenie to generate topologies with different numbers of devices, we introduce a special "Truncate" token and use padding to ensure that all sequences have the same length.

With this customized tokenizer, we pre-train AnalogGenie to **predict the next device pin in the sequence**. Unlike traditional pre-training of LLMs, which often involves randomly cropping sequences from text documents, AnalogGenie's pre-training ensures that each sequence corresponds to a complete circuit topology. Specifically, given an unsupervised corpus of tokens $\mathcal{U} = \{u_1, \ldots, u_n\}$ that represent one circuit topology, we aim to maximize the standard language modeling objective (Radford et al., 2018) to train AnalogGenie. During generation, AnalogGenie begins with a single context token, "VSS" (the starting node for all Eulerian circuits) and completes the rest of the sequence, ensuring it represents a valid circuit topology. The pre-trained AnalogGenie model is capable of learning from various circuit topologies without requiring knowledge of their performance

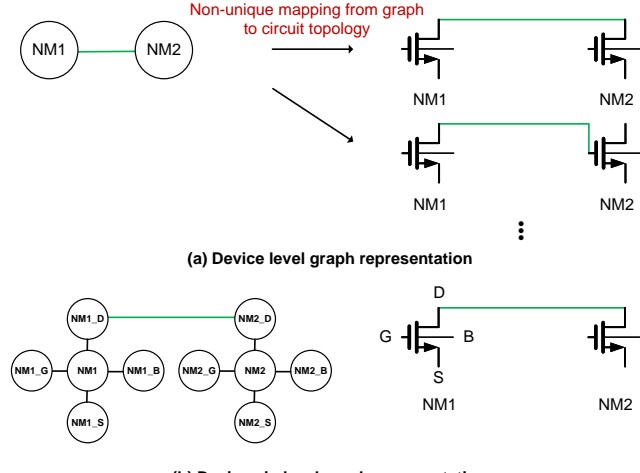

Figure 3: Comparison between two different circuit graph representations. (a) An example showing the limitation of device level graph representation used by previous graph generation work (Dong et al., 2023; Lu et al., 2023), which oversimplified the analog circuit connection and led to non-unique mapping from graph to circuit topology during generation. (b) Our device pin-level graph representation ensures there is a unique mapping between each graph and circuit topology and is able to explicitly represent every connection in a circuit topology.

or specific circuit types. We then can further fine-tune it to target high-performance, unseen circuits for a particular task following the typical manner of reinforcement learning with human feedback.

## 4 RESULTS

### 4.1 EXPERIMENT SETUP

**Dataset and AnalogGenie setup.** Our circuit dataset contains 3350 distinct topologies, spanning 11 circuit types: Op-Amps, LDOs, Bandgap references, Comparators, PLLs, LNAs, PAs, Mixers, VCOs, Power converters, and Switched Capacitor Samplers. The largest circuit comprises 54 devices. The performance of circuits has been evaluated with circuit simulator and thus been labeled. The detailed statistics are shown in Appendix A.1. During training, we first split the topology data set into train and validation sets with a 9 to 1 ratio. Then, we leverage the data augmentation technique in Section 3.3 to generate 70× unique sequences of these circuit topologies. This ensures that all the topologies in the validation set are unseen. Our AnalogGenie model is a decoder-only transformer consisting of 6 hidden layers and 6 attention heads with 11.825 million parameters in total. The vocab size is 1029. The maximum sequence length is 1024.

**Baselines.** We compare AnalogGenie with recent approaches for analog circuit generation, i.e., Ck-tGNN (Dong et al., 2023) based on VAE, and LaMAGIC (Chang et al., 2024) and AnalogCoder (Lai et al., 2024) built on foundation models. These methods are different from AnalogGenie in circuit representation, generation capability and scalability as introduced in Section 2.2. We follow the original work to produce the baseline results.

**Evaluation task and metrics.** We focus on evaluating the generative capabilities of AnalogGenie and baseline models. Specifically, we employ each model to generate topologies for various circuits, including Op-Amps, bandgap references, and power converters, and assess the outcomes based on four key criteria: correctness, scalability, novelty, and performance. (1) **Correctness**: We use a standard circuit simulator to determine whether the generated circuits are simulatable with default sizing, checking for issues such as floating and shorting nodes (i.e., open and short circuits). Circuits that are simulatable are considered valid. (2) **Scalability**: We measure scalability by recording the number of circuit types and the largest valid circuit generated by each model, based on the number of devices. (3) **Novelty**: We assess the novelty of generated topologies by comparing them to existing

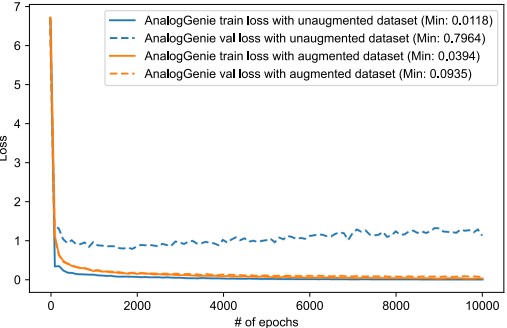

Figure 4: Comparisons between AnalogGenie pretrained with unaugmented data and augmented data. Our augmentation method is able to improve validation loss around 8.5×.

ones in the dataset. A topology is deemed novel if it differs from all known topologies in our datasets. (4) **Performance**: We size each generated topology using a genetic algorithm and use the figure-of-merit (FoM) that considers all major metrics (e.g., gain, bandwidth, power for Op-Amps), as a comprehensive indicator to represent the circuit performance. We compare the best FoM of the circuits generated by each model.

## 4.2 ENHANCED GENERATION CAPABILITIES WITH DATA AUGMENTATION

We begin by examining the impact of the proposed data augmentation technique on the learning capabilities of AnalogGenie. The unaugmented training dataset consists of only 3015 unique sequences due to the train and validation split, with each sequence representing a distinct circuit topology. Figure 4 compares the training performance of AnalogGenie using unaugmented data (3015 sequences) against augmented data (227766 sequences). The results demonstrate that our augmentation technique reduces validation loss by approximately 8.5×. We also evaluate how this improvement influences the quality of circuit generation. The results indicate that without augmentation, AnalogGenie struggled with overfitting, leading to failures in generating valid circuits. In contrast, augmentation significantly enhances generation quality, increasing the number of valid circuits by 73.5× as shown in Table 1.

Table 1: Performance comparison between AnalogGenie and existing analog circuit topology generation work.

| Evaluation Metric | Valid circuits (%) ↑ | Topology scale ↑ | | Novel circuits (%) ↑ | FoM ↑ | | |
|---|---|---|---|---|---|---|---|
| | | Topology type | Number of devices | | Op-Amp | Power Converter | Bandgap |
| CktGNN | 67.5 | 1 | 22 | 93.1 | 10.9 | - | - |
| LaMAGIC | 68.2 | 1 | 4 | 12.7 | - | 2.2 | - |
| AnalogCoder | 57.3 | 7 | 10 | 8.9 | 1.7 | - | - |
| **AnalogGenie (unaug+pretrain)** | **1** | **>11** | **20** | **82.1** | **0** | **0** | **0** |
| **AnalogGenie (aug+pretrain)** | **73.5** | **>11** | **63** | **98.9** | **19.3** | **2.5** | **17.2** |
| **AnalogGenie (aug+pretrain+finetune)** | **93.2** | **>11** | **56** | **99** | **36.5** | **3.3** | **21.9** |

## 4.3 COMPARISONS WITH STATE-OF-THE-ARTS

Next, we compare AnalogGenie with existing VAE model (CktGNN (Dong et al., 2023)) and foundation models (LaMAGIC (Chang et al., 2024) and AnalogCoder (Lai et al., 2024)) addressing the problem of analog circuit topology generation. The comparisons are shown in Table 1.

**Correctness**: AnalogCoder generates only 57.3% valid circuits, primarily due to the error-prone nature of code generation. CktGNN and LaMAGIC, which employ graph generation techniques, achieve slightly better results, with 67.5% and 68.2% valid circuits, respectively. They rely on high-level graph representations, which pose interpretability issues (i.e., ambiguous connections between devices), as discussed in Section 3.1. In contrast, AnalogGenie uses an expressive device pin-level representation, where each connection is explicitly defined, eliminating the mapping issues seen

with the other models. As such, AnalogGenie demonstrates superior correctness, achieving 73.5% valid circuits after pretraining and an impressive 93.2% after fine-tuning.

**Scalability**: CktGNN and LaMAGIC are developed for specific types of circuits, such as Op-Amps or power converters, which limits their ability to generate beyond those particular types. Analog-Coder is able to design 7 circuit types. AnalogGenie demonstrates a zero-shot capability to generate circuit types outside its training set, which includes 11 analog circuit types, as shown in Appendix A.4.2.

Beyond topology types, LaMAGIC is constrained by its graph representation, which contains only four device nodes, preventing it from generating larger circuits. CktGNN has a similar limitation with its fixed-node graph representation, which supports at most 22 devices. AnalogCoder also struggles with scalability, making it capable of generating circuit topologies with a maximum of 10 devices. This restriction arises from its prompt template, which requires users to specify small circuit examples with limited device counts, leading the GPT model to generate circuits of similar size. In contrast, AnalogGenie benefits from a comprehensive dataset, which includes topologies with over 50 devices. Its sequential representation efficiently captures these larger designs, enabling AnalogGenie to generate circuits with up to 64 devices after pretraining and 56 devices after fine-tuning, all within a limited sequence length.

**Novelty (unseen designs)**: AnalogCoder generates only 8.9% novel circuits, as it is designed primarily for task completion rather than exploring new topologies. LaMAGIC, limited to circuits with just four devices, produces 12.7% novel circuits. CktGNN, with its graph model supporting up to 22 device nodes, achieves 93.1% novel circuit discovery. Remarkably, AnalogGenie generates nearly 100% novel circuits, leveraging its ability to design large circuits from scratch. Visualizations of these novel circuits are provided in Appendix A.4.1

**Performance**: For Op-Amp design, AnalogCoder achieves a low FoM of just 1.7, constrained by the limited design options available through its prompt engineering. CktGNN performs significantly better with a FoM of 10.9, though it is still restricted by the lack of detailed low-level design control. AnalogGenie, with its expressive graph structure and flexible bottom-up generation method, initially achieves a FoM of 19 after pre-training. However, following fine-tuning with a focus on high-performance Op-Amp design, AnalogGenie makes impressive gains, reaching a FoM of 36.5. In power converter design, AnalogGenie performs similarly to LaMAGIC, achieving a FoM of 2.5 compared to LaMAGIC's 2.2. After fine-tuning, AnalogGenie further improves, discovering topologies that achieve a FoM of 3.3, thanks to its larger design capacity. Lastly, AnalogGenie stands out as the only model capable of designing bandgap reference circuits. Its pre-trained model achieves a FoM of 17.2, while fine-tuning elevates this to an outstanding FoM of 21.9.

In conclusion, AnalogGenie demonstrates unmatched superiority over AnalogCoder, CktGNN, and LaMAGIC across key metrics such as correctness, scalability, novelty, and performance. This advantage stems from its expressive device pin-level graph representation, which ensures precise and accurate circuit generation, combined with its efficient sequential data structure that supports larger, more complex designs. Additionally, its bottom-up generation approach allows for greater flexibility and innovation in circuit topology, enabling AnalogGenie to explore beyond the limitations faced by the other models. These strengths collectively make AnalogGenie a highly versatile and powerful tool for analog circuit topology design.

## 5 CONCLUSION AND FUTURE WORK

In this paper, we have introduced AnalogGenie, a generative engine built on a GPT model to generate diverse analog circuits by predicting the next device pin to connect within a circuit. AnalogGenie addresses two key gaps in the field: building a comprehensive dataset of analog circuit topology and developing a scalable sequence-based graph representation universal to analog circuits. Experimental results across three types of analog circuit benchmarks demonstrate that our method can efficiently discover previously unseen circuit topologies in a scalable manner. Given the expressive nature of our representation and the bottom-up generative approach, we believe that our method has broader applicability beyond analog circuit topology generation and can be generalized to digital design as well. Ultimately, we consider our work to pave the way for the integration of generative

AI into IC design, fostering a mutually beneficial relationship where IC design enhances generative AI capabilities, while generative AI accelerates IC design advancements.

**Limitations and future work.** AnalogGenie is a comprehensive framework that combines a domain-specific generative engine for discovering analog circuit topologies with a genetic algorithm for optimizing the parameters (e.g., sizing and bias) of the generated topologies. The primary focus of our work is on discovering topologies with a high likelihood of achieving a superior FoM once sized. While the current sizing algorithm is effective, its sample efficiency can be improved by exploring more advanced alternatives (Wang et al., 2020). Additionally, for digital circuit development, we will consider combining AnalogGenie's graph generation approach with code generation work to enhance its ability.

## 6 REPRODUCEBILITY STATEMENT

The main theoretical backbone of our paper is Theorem 3.2.1. We have already shown its proof in Section 3.2. Furthermore, we discussed our experiment setup and implementation details in Section 4.1. We also provide open-source code in supplementary material, including data augmentation, pretraining, and finetuning. The genetic algorithm sizing framework and Ngpsice simulation infrastructure are also provided in the supplementary material. For our open-sourced circuit dataset, we provide its statistics in Appendix A.1. We will make our code and dataset public on Github in the future.

## 7 ACKNOWLEDGEMENTS

This work was partially supported by NSF Award #2416375. We also thank Professor Lei Li of Carnegie Mellon University for valuable discussions on graph generation.

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

# A APPENDIX

## A.1 DATASET STATISTICS

Figure 5 and Figure 6 are the statistics of our open-source dataset. The total number of topologies is 3350, and all of them are unique.

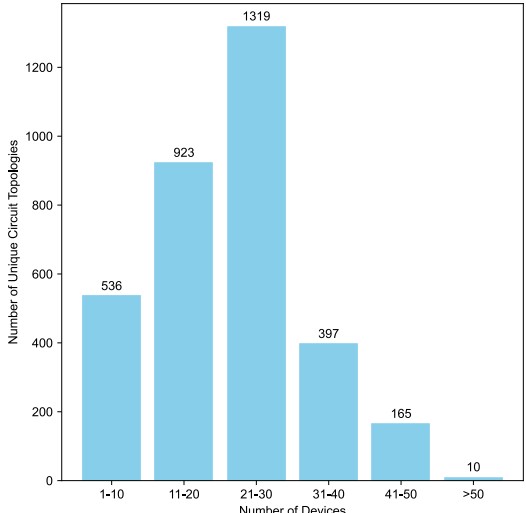

Figure 5: Our analog circuit dataset's device number distribution.

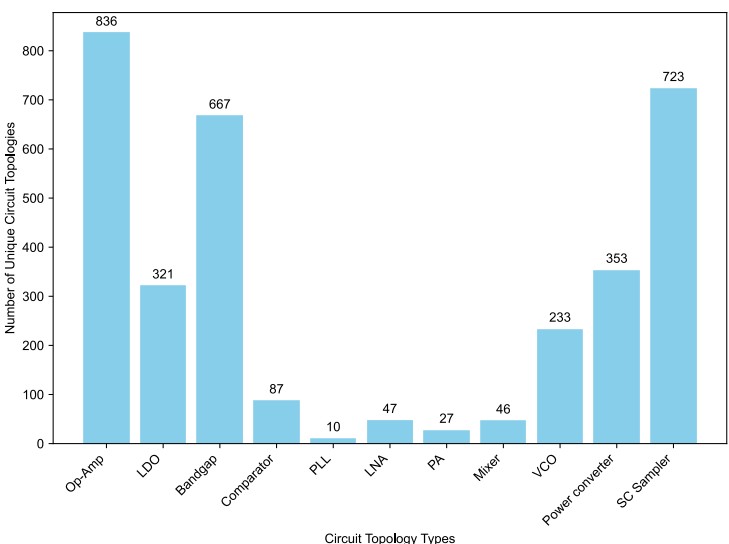

Figure 6: Our analog circuit dataset's circuit topology type distribution.

## A.2 Tokenizer lookup table

Table 2 shows the tokenizer lookup table we used in our experiment. Specifically, our lookup table does not only describe basic devices (e.g., NMOS, PMOS, etc.) but also describes logic gates (e.g., INV, XOR, etc.) that consist of multiple devices so it can scale up to describe large digital circuits for constructing mixed-signal circuits.

Table 2: The tokenizer's look-up table we used in our experiment for device to index mapping

| Device | Index | Device | Index | Device | Index | Device | Index |
|--------|-------|--------|-------|--------|-------|--------|-------|
| NM1 | 0 | NM1_D | 1 | NM1_G | 2 | NM1_S | 3 |
| NM1_B | 4 | NM2 | 5 | ... | ... | NM25_B | 124 |
| PM1 | 125 | PM1_D | 126 | PM1_G | 127 | PM1_S | 128 |
| PM1_B | 129 | PM2 | 130 | ... | ... | PM25_B | 249 |
| NPN1 | 250 | NPN1_C | 251 | NPN1_B | 252 | NPN1_E | 253 |
| NPN2 | 254 | ... | ... | NPN25_E | 349 | PNP1 | 350 |
| PNP1_C | 351 | PNP1_B | 352 | PNP1_E | 353 | PNP2 | 354 |
| ... | ... | PNP25_E | 449 | R1 | 450 | R1_P | 451 |
| R1_N | 452 | R2 | 453 | ... | ... | R25_N | 524 |
| C1 | 525 | C1_P | 526 | C1_N | 527 | C2 | 528 |
| ... | ... | C25_N | 599 | L1 | 600 | L1_P | 601 |
| L1_N | 602 | L2 | 603 | ... | ... | L25_N | 674 |
| DIO1 | 675 | DIO1_P | 676 | DIO1_N | 677 | DIO2 | 678 |
| ... | ... | DIO25_N | 749 | XOR1 | 750 | XOR1_A | 751 |
| XOR1_B | 752 | XOR1_VDD | 753 | XOR1_VSS | 754 | XOR1_Y | 755 |
| XOR2 | 756 | ... | ... | XOR5_Y | 779 | INV1 | 815 |
| INV1_A | 816 | INV1_Q | 817 | INV1_VDD | 818 | INV1_VSS | 819 |
| INV2 | 820 | ... | ... | INV10_VSS | 864 | TG1 | 865 |
| TG1_A | 866 | TG1_B | 867 | TG1_C | 868 | TG1_VDD | 869 |
| TG1_VSS | 870 | TG2 | 871 | ... | ... | TG10_VSS | 924 |
| VIN1 | 925 | VIN2 | 926 | VIN3 | 927 | VIN4 | 928 |
| VIN5 | 929 | IIN1 | 930 | IIN2 | 931 | ... | ... |
| LOGICQB1 | 1024 | LOGICQB2 | 1025 | VDD | 1026 | VSS | 1027 |
| TRUNCATE | 1028 | | | | | | |

A.3  More details about Eulerian circuit and Data augmentation

As shown in Figure 7, since we need to represent a circuit topology in an expressive way (i.e., device pin level), a small circuit with two devices can lead to a graph with 14 nodes. Using an adjacency matrix to represent it will take $14 \times 14 = 256$ elements. On the other hand, the Eulerian circuit only needs 43 elements, which is around $5.95\times$ smaller than the adjacency matrix representation. For our data augmentation, we permutate how the DFS algorithm explores its neighbor to generate unique Eulerian circuits. The number of Eulerian circuits will drastically increase with the number of devices.

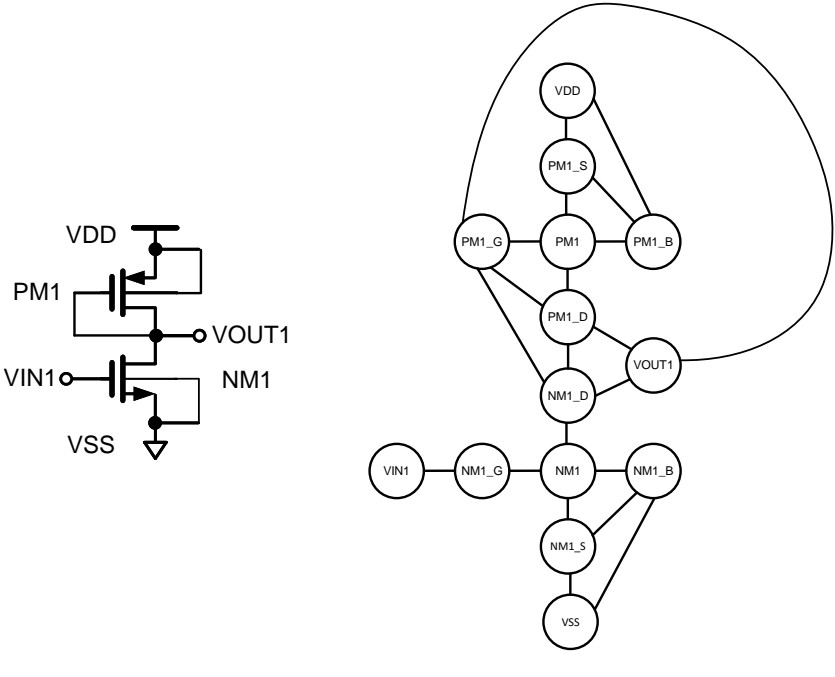

**(a) Topology**        **(b) Device pin level graph representation**

['VSS' 'NM1_S' 'NM1' 'NM1_D' 'VOUT1' 'PM1_D' 'PM1' 'PM1_G' 'VOUT1''PM1_G' 'PM1_D' 'NM1_D' 'PM1_G' 'NM1_D'
'PM1_D' 'PM1_G' 'PM1' 'PM1_S' 'VDD' 'PM1_B' 'PM1' 'PM1_B' 'PM1_S' 'PM1_B' 'VDD' 'PM1_S' 'PM1' 'PM1_D' 'VOUT1'
'NM1_D' 'NM1' 'NM1_G' 'VIN1' 'NM1_G' 'NM1' 'NM1_B' 'VSS' 'NM1_B' 'NM1_S' 'NM1_B' 'NM1' 'NM1_S' 'VSS']

['VSS' 'NM1_B' 'NM1' 'NM1_D' 'VOUT1' 'PM1_D' 'PM1' 'PM1_G' 'VOUT1''PM1_G' 'PM1_D' 'NM1_D' 'PM1_G' 'NM1_D'
'PM1_D' 'PM1_G' 'PM1' 'PM1_S' 'VDD' 'PM1_B' 'PM1' 'PM1_B' 'PM1_S' 'PM1_B' 'VDD' 'PM1_S' 'PM1' 'PM1_D' 'VOUT1'
'NM1_D' 'NM1' 'NM1_G' 'VIN1' 'NM1_G' 'NM1' 'NM1_S' 'VSS' 'NM1_S' 'NM1_B' 'NM1_S' 'NM1' 'NM1_B' 'VSS']

['VSS' 'NM1_S' 'NM1_B' 'VSS' 'NM1_B' 'NM1' 'NM1_D' 'VOUT1' 'PM1_D' 'PM1' 'PM1_G' 'VOUT1' 'PM1_G' 'PM1_D'
'NM1_D' 'PM1_G' 'NM1_D' 'PM1_D' 'PM1_G' 'PM1' 'PM1_S' 'VDD' 'PM1_B' 'PM1' 'PM1_B' 'PM1_S' 'PM1_B' 'VDD'
'PM1_S' 'PM1' 'PM1_D' 'VOUT1' 'NM1_D' 'NM1' 'NM1_G' 'VIN1' 'NM1_G' 'NM1' 'NM1_S' 'NM1' 'NM1_B' 'NM1_S' 'VSS']

['VSS' 'NM1_B' 'NM1_S' 'VSS' 'NM1_S' 'NM1' 'NM1_D' 'VOUT1' 'PM1_D' 'PM1' 'PM1_G' 'VOUT1' 'PM1_G' 'PM1_D'
'NM1_D' 'PM1_G' 'NM1_D' 'PM1_D' 'PM1_G' 'PM1' 'PM1_S' 'VDD' 'PM1_B' 'PM1' 'PM1_B' 'PM1_S' 'PM1_B' 'VDD'
'PM1_S' 'PM1' 'PM1_D' 'VOUT1' 'NM1_D' 'NM1' 'NM1_G' 'VIN1' 'NM1_G' 'NM1' 'NM1_B' 'NM1' 'NM1_S' 'NM1_B' 'VSS']

**(C) Four Eulerian circuits we found by using DFS**

Figure 7: An example of analog circuit topology, its device pin level graph representation, four unique Eulerian circuits we found by using DFS.

## A.4 ANALOGGENIE'S GENERATED CIRCUIT TOPOLOGY VISUALIZATION

In this section, we show some of the novel circuits AnalogGenie generated and demonstrate its zero-shot capability by generating circuits that belong to a type that is not included in the dataset. Particularly, to visualize the circuit schematic, we manually draw all the examples in Cadence Virtuoso, which is an industry-standard analog schematic edit tool.

### A.4.1 NOVEL CIRCUITS

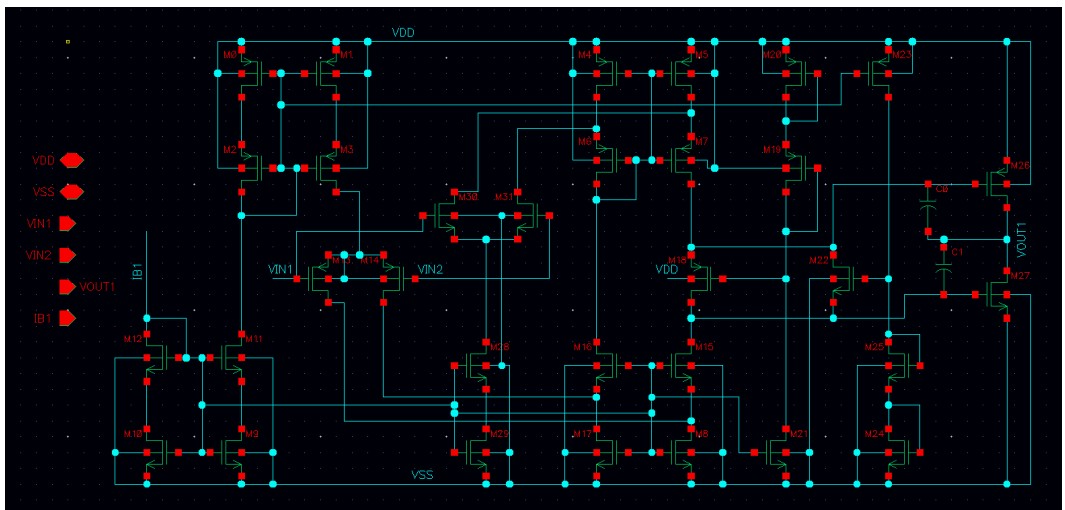

Figure 8: A novel Op-Amp circuit generated by AnalogGenie with GBW = 12 MHz, $C_L$ = 100 pF, Power = 32.88 mW, and FoM = 36.5.

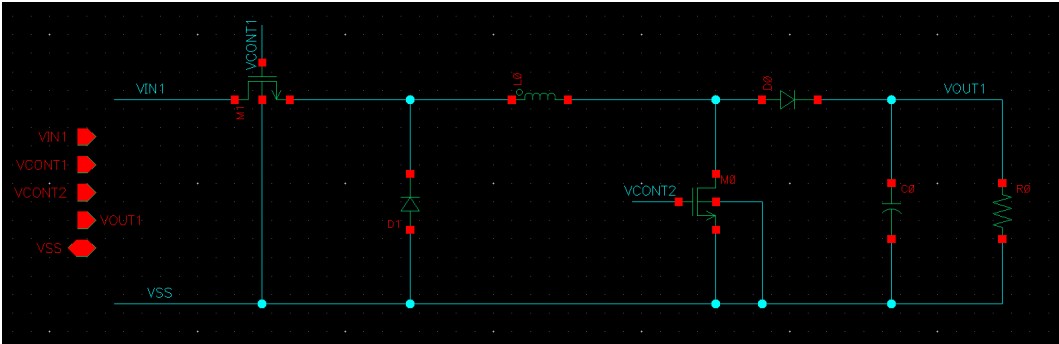

Figure 9: A novel DC converter circuit generated by AnalogGenie with Efficiency = 0.95, Voltage conversion ratio = 2.35, and FoM = 3.3.

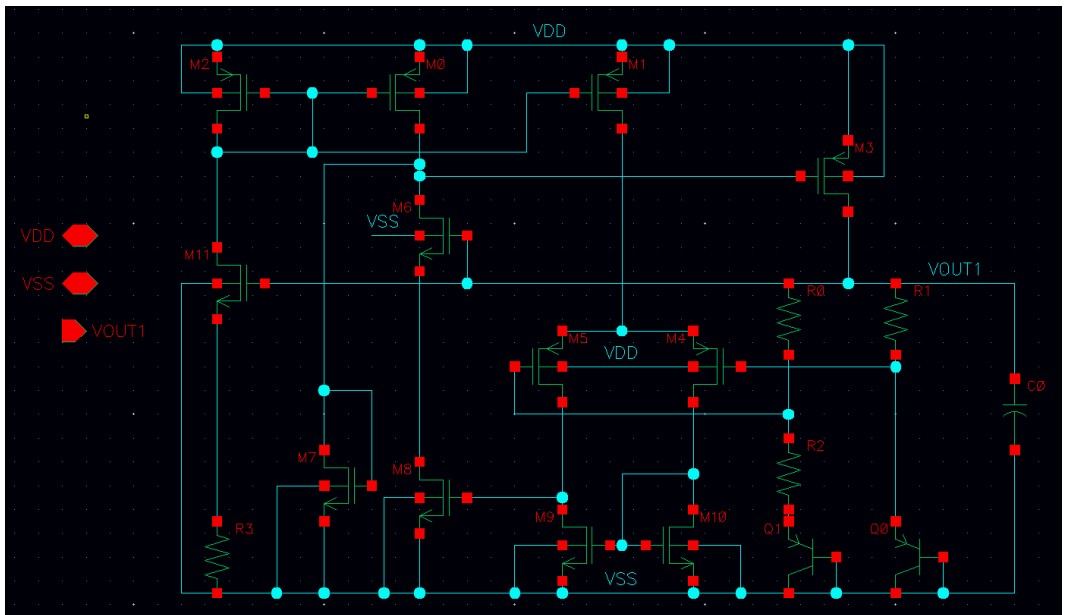

Figure 10: A novel bandgap reference circuit generated by AnalogGenie with TC = 3 ppm/°C, Line regulation = 0.196 %/V, PSRR = 70 dB, and FoM = 21.9.

### A.4.2 ZERO-SHOT GENERATION

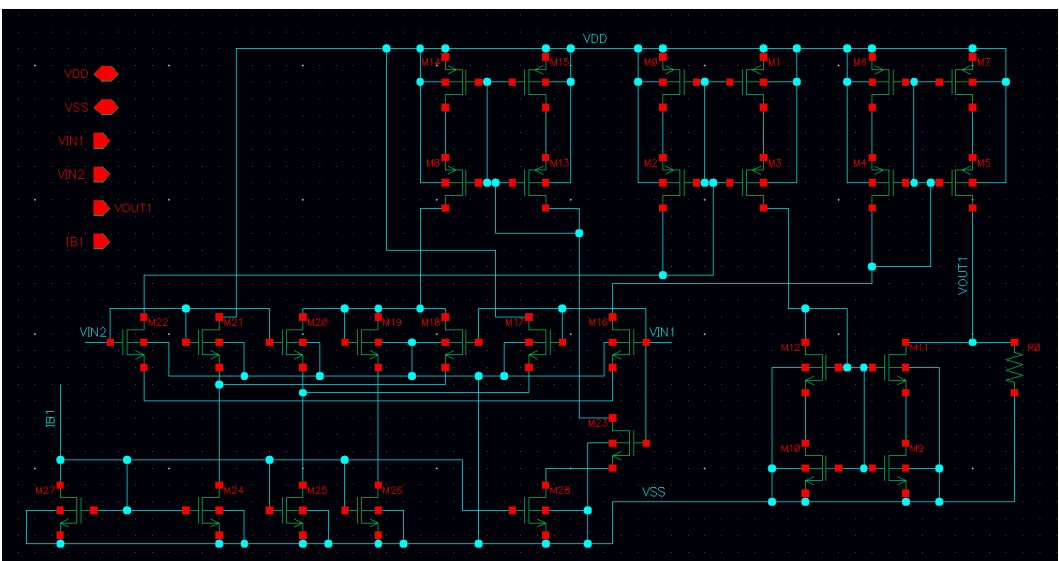

Figure 11: An example showing AnalogGenie's zero-shot ability by generating a transconductance amplifier which is a circuit type that is not included in our dataset.

A.5    FAILED EXAMPLES GENERATED BY ANALOGGENIE PRE-TRAINED MODEL

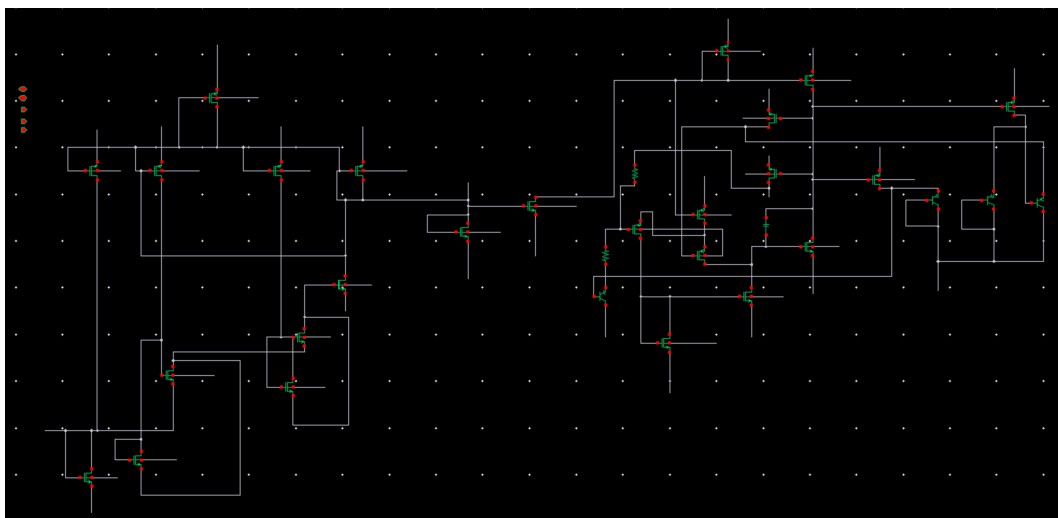

Figure 12: Failed example 1.

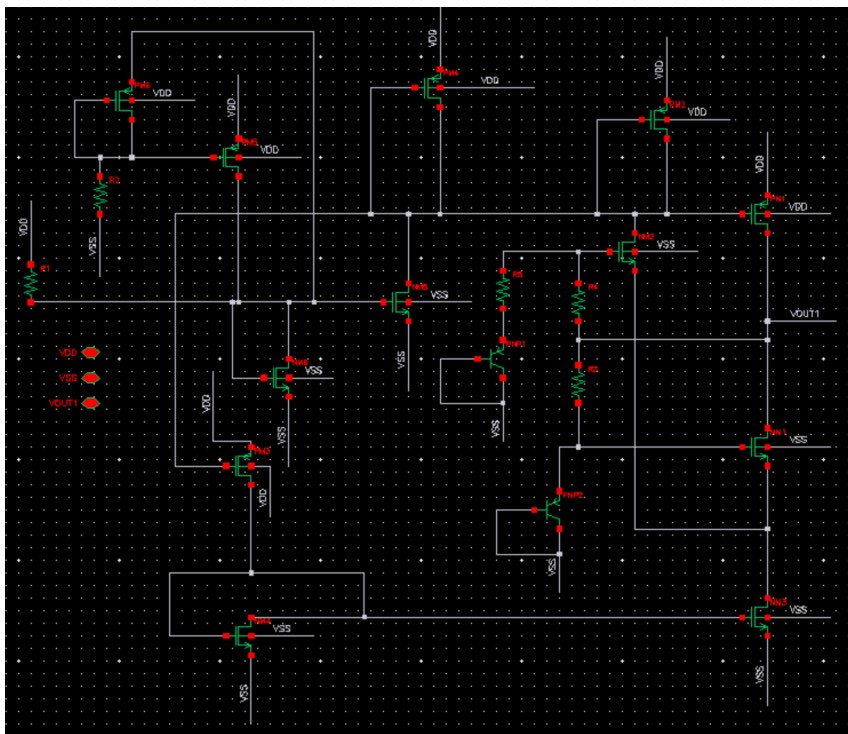

Figure 13: Failed example 2.

