# OpenReview forum: "AnalogGenie: A Generative Engine for Automatic Discovery of Analog Circuit Topologies"
_ICLR.cc/2025/Conference — ICLR 2025 Spotlight_

### Official Review · Reviewer_nrPJ · 2024-10-28

**Soundness:** 4
**Presentation:** 4
**Contribution:** 4
**Rating:** 8
**Confidence:** 5

**Summary:**

This paper makes a substantial contribution to the automation of analog integrated circuit design, an enduringly challenging problem in electronic design automation (EDA). It introduces a novel approach, establishing a unique, one-to-one mapping between a graph and the circuit topology. Here, each circuit connection is explicitly represented, requiring analog circuits to be modeled at the pin level. A key innovation is a method to transform the undirected graph into a sequence format. Following data augmentation, the authors train a GPT model that generates diverse analog circuits by predicting the next device pin connection within a circuit. The evaluation results demonstrate that the proposed GPT model significantly outperforms previous models.

The work is impressive, presenting AnalogGenie, a powerful GPT-based model capable of generating a much broader range of large-scale, valid, unseen, and high-performance analog circuit topologies than prior models. It also introduces an efficient, sequence-based, pin-level graph representation that effectively captures complex analog circuit structures. Additionally, the paper constructs a comprehensive, large-scale, high-quality dataset of analog circuit topologies, covering nearly all major categories, including RF circuits. This dataset will likely serve as a valuable resource for further advancements in analog circuit design automation.

**Strengths:**

This is a remarkable work with several innovative contributions. First, it introduces an efficient and precise pin-level graph representation of analog topologies, enabling a unique one-to-one mapping between the graph and the circuit topology. Second, the authors treat undirected edges as two directed arcs in opposite directions, ensuring that the graph contains at least one Eulerian circuit starting from any vertex. Using this approach, they can sequentialize the graph of an analog circuit into an Eulerian circuit—a trail that visits each edge exactly once and both starts and ends at the "VSS" node. Third, the work implements data augmentation by enumerating all possible trails, effectively reducing overfitting and enhancing the model's generalization ability. Fourth, by creatively applying techniques from natural language processing, the paper reframes analog topology generation as a next-device-pin prediction task, with "VSS" acting as the starting token.

Additionally, this work provides a comprehensive, high-quality, large-scale dataset, poised to greatly advance analog design automation. Thanks to these innovative aspects, AnalogGenie achieves state-of-the-art performance. As a reviewer with experience in analog circuit design during undergraduate studies, I find the circuits presented in the Appendix valuable and practically significant.

**Weaknesses:**

The dataset introduced in this work encompasses various types of analog circuits, though the experiments conducted focus primarily on commonly used analog circuits. While these circuit types—such as operational amplifiers, power converters, and bandgap references—are indeed foundational in analog electronics, I hope the authors will consider expanding their experiments to include additional analog circuit types in future work. Furthermore, I encourage the authors to present these well-designed circuits to senior engineers for evaluation and, if possible, pursue a tape-out to demonstrate real-world feasibility.

**Questions:**

1.The authors are committed to open-sourcing this, correct? It would be valuable if they honor that commitment, as their work, if as described, could greatly benefit the automation of analog circuit design.?
2.Can the authors explain how the sizing is done?

---

> ### Author Response · Authors · 2024-11-22
> **Author Response 1/1**
>
> We appreciate the reviewer’s constructive feedbacks and positive review.
>
> > **Q1:** The dataset introduced in this work encompasses various types of analog circuits, though the experiments conducted focus primarily on commonly used analog circuits. While these circuit types—such as operational amplifiers, power converters, and bandgap references—are indeed foundational in analog electronics, I hope the authors will consider expanding their experiments to include additional analog circuit types in future work. Furthermore, I encourage the authors to present these well-designed circuits to senior engineers for evaluation and, if possible, pursue a tape-out to demonstrate real-world feasibility.
>
> We appreciate your recognition of the foundational circuit types we focused on in this work. Our goal is to make this dataset sustainable, as we aim to demonstrate the versatility and generalizability of our approach. Thus, we will continuously add more analog circuits to the dataset to make it an essential infrastructure in analog design automation. Regarding your suggestion to involve senior engineers for evaluation, we agree that expert feedback would enhance the practical relevance of our work. This is a logical next step, and we are already planning collaborations to assess the generated circuits more rigorously. Finally, pursuing a tape-out is an excellent idea to validate the real-world feasibility of the circuits. While this is beyond the scope of the current study, it aligns with our long-term vision for silicon implementation, and we are actively exploring opportunities to realize this. Thank you again for your insights—they will help us strengthen and broaden the impact of our research.
>
> > **Q2:** The authors are committed to open-sourcing this, correct? It would be valuable if they honor that commitment, as their work, if as described, could greatly benefit the automation of analog circuit design.?
>
> Yes, we will open-source our code and dataset. As mentioned in **Section 6 Reproducibility Statement**, we will make our code and dataset public on Github in the future.
>
> > **Q3:** Can the authors explain how the sizing is done?
>
> As we written in **Section 4.1**, we size each generated topology using a genetic algorithm (GA) and use the figure-of-merit (FoM) that considers all major metrics (e.g., gain, bandwidth, power for Op-Amps), as a comprehensive indicator to represent the circuit performance. The simulator is Ngspice, and the technology is BSIM4 45nm. The genetic algorithm formulates the sizing problem as a black box optimization problem aiming to find the candidate solution that has the highest FoM. It iteratively applies selection, crossover, and mutation to explore and exploit the search space (i.e., the range of sizing and bias we allow GA to tune) and terminates the optimization when it exhausts the entire simulation budget. We selected the genetic algorithm as our sizing algorithm due to its model-free simplicity and we plan to explore more advanced sizing algorithms in future work [1-3].
>
>
>
> [1] Budak, Ahmet F., et al. "Dnn-opt: An rl inspired optimization for analog circuit sizing using deep neural networks." *2021 58th ACM/IEEE Design Automation Conference (DAC)*. IEEE, 2021.
>
> [2] Wang, Hanrui, et al. "GCN-RL circuit designer: Transferable transistor sizing with graph neural networks and reinforcement learning." *2020 57th ACM/IEEE Design Automation Conference (DAC)*. IEEE, 2020.
>
> [3] Lyu, Wenlong, et al. "An efficient bayesian optimization approach for automated optimization of analog circuits." *IEEE Transactions on Circuits and Systems I: Regular Papers* 65.6 (2017): 1954-1967.

---

### Official Review · Reviewer_KSXW · 2024-11-01

**Soundness:** 3
**Presentation:** 3
**Contribution:** 3
**Rating:** 8
**Confidence:** 3

**Summary:**

This paper introduces "AnalogGenie," a new tool designed to automatically discover analog circuit topologies. The authors aim to tackle some of the biggest hurdles in automating analog IC design, like the lack of comprehensive datasets and effective ways to represent analog circuits.

The main focus here is on automating the design of analog ICs, which is notoriously tricky. Unlike digital ICs, analog circuits don't have a straightforward hierarchical structure and rely heavily on designer intuition and experience. This makes automation a real challenge, and that's exactly what this paper tries to address.

**Strengths:**

The authors reference several earlier efforts, like CktGNN, LaMAGIC, and AnalogCoder, that have explored using generative AI for analog circuit design. These prior works show promise, but there's still a lot of unexplored potential in this area.

AnalogGenie takes a novel approach by creating a large dataset of over 3,000 analog circuit topologies and developing a scalable sequence-based graph representation. They also introduce a customized tokenizer and a pre-training method that allows the model to learn from these topologies without needing to know specific performance metrics or circuit types.

While the paper doesn't dive into the nitty-gritty of the experiments, it does claim that AnalogGenie can produce a wide range of large-scale, valid, and high-performance analog circuit topologies, outperforming previous methods.

Overall, AnalogGenie seems like a promising step forward in the automation of analog circuit design. However, it would be great to see more detailed discussions on its limitations and how it stacks up against other state-of-the-art methods. More insights on future directions could also help guide further research in this exciting area.

**Weaknesses:**

One downside is that the paper doesn't really discuss any limitations of AnalogGenie or suggest areas for future research. It hints at a lot of untapped potential in the field of analog IC design automation and suggests that AnalogGenie's approach might even be applicable to digital design.

**Questions:**

The paper mentions the use of a customized tokenizer and a finetuning process for AnalogGenie. How do they help tailor AnalogGenie to the analog circuit design task and improve the generation quality? Additionally, will this custom tokenizer affect its performance on other tasks, potentially reducing the inherent reasoning ability of the language model? Are there any metrics or experimental methods that can be used to verify these issues?

---

> ### Author Response · Authors · 2024-11-22
> **Author Response 1/1**
>
> We appreciate the reviewer’s positive feedback and thoughtful comments.
>
> > **Q1:** One downside is that the paper doesn't really discuss any limitations of AnalogGenie or suggest areas for future research. It hints at a lot of untapped potential in the field of analog IC design automation and suggests that AnalogGenie's approach might even be applicable to digital design.
>
> We have added limitations and future work discussion in **Section 5 Conclusion and Future Work** of our paper revision.
>
> "AnalogGenie is a comprehensive framework that combines a domain-specific generative engine for discovering analog circuit topologies with a genetic algorithm for optimizing the parameters (e.g., sizing and bias) of the generated topologies. The primary focus of our work is on discovering topologies with a high likelihood of achieving a superior FoM once sized. While the current sizing algorithm is effective, its sample efficiency can be improved by exploring more advanced alternatives [1]. Additionally, for digital circuit development, we will consider combining AnalogGenie's graph generation approach with code generation work to enhance its ability."
>
> > **Q2:** The paper mentions the use of a customized tokenizer and a finetuning process for AnalogGenie. How do they help tailor AnalogGenie to the analog circuit design task and improve the generation quality?
>
> Our customized tokenizer uses design pins as tokens, directly aligning the language model’s atomic behavior (predicting the next token in context) with the analog circuit design process (connecting the next device pin based on the existing circuit). **This alignment improves the correctness of circuit generation compared to approaches that rely on natural language tokenizers to describe circuits at a high level.** For instance, in our comparison between AnalogGenie and AnalogCoder (Table 1), which uses a typical language tokenizer, AnalogCoder produces only 57.3% valid circuits, largely due to the error-prone nature of code generation. In contrast,  AnalogGenie achieves 73.5% valid circuits with only pretraining and improves further to 93.2% valid circuits after fine-tuning, demonstrating the efficacy of its customized tokenizer.
>
> **A pre-trained model lacks inherent preferences for circuit types or performance metrics, making the generation random and chaotic.** To address this, we utilize **reinforcement learning with human feedback (RLHF) to fine-tune the pre-trained model, aligning its outputs with human-defined preferences for circuit type and performance.** The finetuning begins by training a reward model that evaluates newly generated topologies by assigning a score in terms of correctness, novelty, and performance. To train this model, we initially label a limited set of circuit topologies based on their type and performance using a genetic algorithm for sizing and performing SPICE simulation. Once the reward model is trained, we use **proximal policy optimization (PPO)** [2] to fine-tune AnalogGenie. In each epoch, the pre-trained model generates a batch of new topologies, and the reward model evaluates and scores them. PPO is then applied to optimize the model, maximizing the expected accumulated reward scores. After fine-tuning, AnalogGenie demonstrates the ability to discover high-performance, novel, and valid circuits as shown in Table 1.
>
> > **Q3:** Additionally, will this custom tokenizer affect its performance on other tasks, potentially reducing the inherent reasoning ability of the language model? Are there any metrics or experimental methods that can be used to verify these issues?
>
> Our model is a domain-specific circuit generation model with custom tokenizer, repurposing the existing tokens from text generation to circuit generation. In other words, it is not applicable to natural language generation tasks as typical LLMs. Thus, studying its reasoning ability under the context of LLM shows no practical meaning.
>
>
>
> [1] Wang, Hanrui, et al. "GCN-RL circuit designer: Transferable transistor sizing with graph neural networks and reinforcement learning." *2020 57th ACM/IEEE Design Automation Conference (DAC)*. IEEE, 2020.
>
> [2] Ouyang, Long, et al. "Training language models to follow instructions with human feedback." *Advances in neural information processing systems* 35 (2022): 27730-27744.

---

> > ### Comment · Reviewer_KSXW · 2024-11-26
> >
> > Thank you for the detailed responses. The added discussion on limitations and future work is helpful, and the explanation of the tokenizer and fine-tuning process clarifies their impact on generation quality. The clarification about the model's domain-specific focus also makes sense. Everything looks good to me.

---

> > > ### Author Response · Authors · 2024-11-26
> > >
> > > Thanks for your positive feedback. Feel free to ask us any further questions.

---

### Official Review · Reviewer_U6NJ · 2024-11-01

**Soundness:** 3
**Presentation:** 3
**Contribution:** 3
**Rating:** 6
**Confidence:** 3

**Summary:**

The authors introduce a novel methodology for analog circuit design, leveraging the capabilities of generative AI (specifically, GPT-based models) alongside a fine-grained Directed Acyclic Graph (DAG) representation. Additionally, they present a new dataset tailored to this task, developed through data augmentation techniques.

**Strengths:**

1. The authors propose an innovative RAG representation for analog circuits, where each node corresponds to a device pin. This approach enables the model to capture low-level features and generate topologies in a fine-grained way.

2. The authors present a new open-source dataset that comprises LDOs, bandgap references, comparators, PLLs, LNAs, PAs, mixers, and VCOs. Unlike previous datasets, this dataset includes performance metrics for each circuit. To ensure accuracy, the researchers manually implemented each circuit in an industry-standard design tool and conducted simulations, which is a significant manual effort.

3. The authors avoid using adjacency matrices to represent circuit topologies. Instead, they employ an Eulerian graph representation, which effectively reduces the model’s memory requirements.

4. To the best of my knowledge, this is the first work to apply generative AI models to tasks in analog circuit design.

5. The authors employ data augmentation techniques by learning multiple unique Eulerian circuits that represent the same topology. This approach mitigates overfitting and reduces inductive bias, enhancing the model's generalization capability.

**Weaknesses:**

The experimental part over-emphasized the performance of the model itself. There is no data that shows the performance of the circuits generated by the AGI models. Since the generated circuits are about to be used in the real physical world, I think it is essential to analyze some basic and widely-used analog circuits like OpAmps(including the slew rate, GBW, CMRR, AoI, Ib, Vos, etc.), LDO(including Dropout Voltage, PSRR, etc.)

**Questions:**

1. I am quite confused about the expression of inductive bias in graph-based data. In the paper, the authors pointed out that to address the inductive bias issue, the model learned multiple unique Eulerian circuits that represent the same topology. Do these Eulerian circuits have different or share the same inductive bias? Since I am not a researcher in graph neural networks, I hope the authors could give me detailed and easy-to-understand instructions to explain this issue (better include proofs)

2. To to best of my knowledge, the authors use pin-grained graph representation to describe a typical circuit design. Is this strategy a kind of data augmentation since a graph with more nodes can represent more information? If so, the main innovative point of this work is to change the representation to a lower level. If we apply such a strategy to other methods without using AGI models, will those older methods (Like GNN-based models) perform better as well?

3. In the real world, the demand for RF circuits is high and those devices require very strict performance and electrical parameters (like filters, PLL, etc.). Is this system fully qualified to handle such devices' circuit generation? If so, are the generated circuits' performance as good as or even better than those that already exist?

---

> ### Author Response · Authors · 2024-11-22
> **Author Response 1/3**
>
> Thanks for the helpful comments. We address your concerns below.
>
> > **Q1:** The experimental part over-emphasized the performance of the model itself. There is no data that shows the performance of the circuits generated by the AGI models. Since the generated circuits are about to be used in the real physical world, I think it is essential to analyze some basic and widely-used analog circuits like OpAmps(including the slew rate, GBW, CMRR, AoI, Ib, Vos, etc.), LDO(including Dropout Voltage, PSRR, etc.)
>
> **We have shown some real-world circuits generated by AnalogGenie in Appendix 4. The FoM results presented in Table 1 serve as an indicator of the real-world performance of the generated circuits.** As described in **Section 4.1 (Experiment Setup)**, we size each generated topology using a genetic algorithm, employing FoM as a comprehensive performance metric. The FoM incorporates key parameters such as gain, bandwidth, and power for operational amplifiers, ensuring a holistic evaluation of circuit performance. Our simulations are conducted using Ngspice, and the technology node is BSIM4 45nm. To promote reproducibility, the entire sizing framework is open-sourced and available in the supplementary material.
>
> Additionally, in Appendix 4, we provide visualizations and detailed performance metrics for some circuits generated by AnalogGenie, including:
>
> - **Operational Amplifier**: GBW = 12 MHz, CL = 100 pF, Power = 32.88 mW, FoM = 36.5
> - **DC Converter**: Efficiency = 0.95, Voltage Conversion Ratio = 2.35, FoM = 3.3
> - **Bandgap Reference**: TC = 3 ppm/°C, Line Regulation = 0.196%/V, PSRR = 70 dB, FoM = 21.9

---

> ### Author Response · Authors · 2024-11-22
> **Author Response 2/3**
>
> > **Q2:** I am quite confused about the expression of inductive bias in graph-based data. In the paper, the authors pointed out that to address the inductive bias issue, the model learned multiple unique Eulerian circuits that represent the same topology. Do these Eulerian circuits have different or share the same inductive bias? Since I am not a researcher in graph neural networks, I hope the authors could give me detailed and easy-to-understand instructions to explain this issue (better include proofs)
>
> To clarify your question on inductive bias in graph-based data and support our Eulerian circuit representation, we provide detailed explanations and steps taken to address this issue in **Section 3.3**. Permutation invariance is a well-known inductive bias in graph-structured data. For a graph with $n$ nodes, there can be up to $n$ ! equivalent adjacency matrices representing the same graph due to different node orderings. This challenge is particularly relevant to analog circuit graphs, where, for example, multiple NMOS transistors can appear in arbitrary orders (e.g., determining which is labeled NMOS1). **This non-unique representation is a key source of inductive bias**. A robust graph generative model must **assign equal probabilities to all equivalent representations of the same graph.**
>
> AnalogGenie addresses inductive bias at two levels: graph-level and sequence-level:
>
> Inductive bias at the graph level arises primarily from node ordering (e.g., which node is labeled as 1). To ensure a consistent representation, AnalogGenie employs a deterministic breadth-first search (BFS) function to enforce node ordering. This guarantees that each circuit topology has a unique graph representation. More formally, An undirected graph $G=(V, E)$ is defined by its node set $V=\\{v_1, \ldots, v_n\\}$ and edge set $E=\\{(v_i, v_j) \mid v_i, v_j \in V\\}$. One common way to represent a graph is using an adjacency matrix, which requires a node ordering $\pi$ that maps nodes to rows/columns of the adjacency matrix. More precisely, $\pi$ is a permutation function over $V$ (i.e., $\left(\pi\left(v_1\right), \ldots, \pi\left(v_n\right)\right)$ is a permutation of $\left.\left(v_1, \ldots, v_n\right)\right)$. BFS function takes a random permutation $\pi$ as input, picks $\pi\left(v_1\right)$ as the starting node and appends the neighbors of a node into the BFS queue in the order defined by $\pi$. **Note that the BFS function is many-to-one, i.e., multiple permutations can map to the same ordering after applying the BFS function [1].** By using a deterministic BFS function, AnalogGenie ensures that each circuit topology has a unique graph representation.
>
> Even with a unique graph representation, the corresponding sequential representation remains non-unique due to:
>
> - Selection of the starting node for traversal,
> - Whether edges are traversed multiple times, and
> - How the sequence traverses the graph.
>
> To address sequence-level inductive bias, AnalogGenie manually defines the starting node for all sequences to be "VSS" (the starting node for all Eulerian circuits). By utilizing Eulerian circuits, AnalogGenie ensures that our sequence only traverses each directed edge once. To reduce the inductive bias caused by graph traversal, AnalogGenie generates all potential traversal paths through augmentation by using a depth-first search (DFS) algorithm that always starts from "VSS" and permutes how it explores neighboring nodes in each iteration. This approach generates multiple unique Eulerian circuits that represent the same circuit topology, **allowing the model to assign the same probability of generation for each Eulerian circuit without any inductive bias toward a certain representation (i.e., there is no inductive bias anymore) after training.**

---

> ### Author Response · Authors · 2024-11-22
> **Author Response 3/3**
>
> > **Q3:** To to best of my knowledge, the authors use pin-grained graph representation to describe a typical circuit design. Is this strategy a kind of data augmentation since a graph with more nodes can represent more information? If so, the main innovative point of this work is to change the representation to a lower level. If we apply such a strategy to other methods without using AGI models, will those older methods (Like GNN-based models) perform better as well?
>
> Our pin-level graph representation primarily addresses the issue of **avoiding ambiguous generation** by explicitly representing each device pin as a distinct graph node, **which is not for data augmentation.** As discussed in **Section 3.1 and Figure 2**, prior works such as CktGNN and LaMAGIC rely on high-level graph representations, where each node corresponds to an entire device or subgraph. While this abstraction simplifies representation, it omits critical low-level device details, resulting in **ambiguous generation—a single generated graph can be interpreted as multiple distinct topologies.** For example, consider an NMOS transistor (NM) with four pins: drain (D), gate (G), source (S), and body (B). High-level representations abstract the entire NMOS transistor into a single node, making it unclear which pin a connecting edge corresponds to. To ensure a **unique one-to-one mapping** between the graph and the circuit topology, we explicitly represent each device pin as a separate graph node. This approach captures every circuit connection precisely and avoids ambiguity in graph interpretation.
>
> **Our augmentation method involves generating multiple unique sequences for the same graph by permuting the traversal order during depth-first search (DFS).** This augmentation allows the model to learn **diverse schematic generation options**, effectively simulating how a designer might draw the same circuit differently starting from the same entry point (e.g., from "VSS"). In analog circuit topologies, each device pin has multiple potential connection options in real-world scenarios, with no strict rules governing the connection order. Our augmentation ensures that AnalogGenie learns this diversity by providing multiple traversal paths through the graph. This not only improves the model's ability to generalize but also enhances its robustness in generating complex and varied circuit topologies.
>
> We believe that our pin-level graph representation and augmentation approach could be generalized to older models that combine GNNs with transformer architectures, such as Graphormer [2]. These models could benefit from our representation's explicitness and augmentation method's diversity to reduce inductive bias and prevent overfitting. However, exploring such architectures lies beyond the scope of this paper. We plan to investigate it in future work.
>
> > **Q4:** In the real world, the demand for RF circuits is high and those devices require very strict performance and electrical parameters (like filters, PLL, etc.). Is this system fully qualified to handle such devices' circuit generation? If so, are the generated circuits' performance as good as or even better than those that already exist?
>
> **AnalogGenie is capable of discovering RF circuits such as PLLs, LNAs, Mixers, and VCOs, as these are included in the datasets we utilized.** The circuits generated by AnalogGenie demonstrate excellent performance at the schematic level. However, we acknowledge that their post-layout and chip-level performance remains uncertain. Given the high sensitivity of RF circuits to post-layout parasitics, it is challenging to determine whether these generated circuits outperform existing designs without further validation. In future work, we plan to address this by constructing layouts and conducting tape-out experiments to evaluate their real-world performance comprehensively.
>
>
>
> [1] You, Jiaxuan, et al. "Graphrnn: Generating realistic graphs with deep auto-regressive models." *International conference on machine learning*. PMLR, 2018.
>
> [2] Ying, Chengxuan, et al. "Do transformers really perform badly for graph representation?." *Advances in neural information processing systems* 34 (2021): 28877-28888.

---

> > ### Comment · Reviewer_U6NJ · 2024-11-25
> >
> > I sincerely appreciate the author's response, it seems to be good work. However, I need to clarify again that I am not an expert in this field, so I decided to keep the original score for fairness.

---

> > > ### Author Response · Authors · 2024-11-25
> > >
> > > Thanks for recognizing our work. Feel free to let us know if you have any questions.

---

### Official Review · Reviewer_sJmh · 2024-11-04

**Soundness:** 3
**Presentation:** 3
**Contribution:** 4
**Rating:** 8
**Confidence:** 4

**Summary:**

The paper introduces AnalogGenie, a generative tool to create analog circuit topology based on a generative model. The method has two contributions: first, the paper collects a large and complete analog circuit dataset; second, the scalable generation of analog circuits; the paper proposes to transfer the analog circuit into a sequence-based graph, thus enabling the usage of a generative model with token-based generation. With dataset augmentation and pre-training, AnalogGenie can finish 73.5% of problems, outperforming the best baseline with 68.2%.

**Strengths:**

Strength:
* The idea of transferring the analog circuit into a sequence-based graph representing is a novel and interesting idea, enabling the use of a generative model for scalable circuit generation.
* The collection of large-scale analog circuits. It would be great if the authors can release the dataset.

**Weaknesses:**

Weakness:
* The main weakness of this paper is that it only considers generating functional circuits, which limits its effectiveness.  In real-world tasks, analog designers need to optimize the topology based on the design targets, e.g., sizing. Does this work consider performance optimization during the generation process, or is it only considered functional? Have you performed sizing?

More weaknesses please check the questions:

**Questions:**

Questions:
* To ensure the practicality of the designed circuit in real-world circuit design scenarios, the process technology of circuit fabrication should be taken into consideration. How do you think this current work can be extended to incorporate advanced process technologies and further optimize the design for improved performance and efficiency?
* Based on my understanding, the paper didn't propose methods to optimize circuits's FoM. Therefore, the paper shouldn't compare the FoM, as the benefits may come from your large dataset, and the model may reuse some circuits. This is unfair as previous methods cannot access the large dataset you have collected.
* The paper tries to tackle a simple task with topology generation, but how can the more critical analog circuit structure selection and sizing problem be solved?
* For fine-tuning, how do you ensure the validation circuit has no overlapped knowledge in the fine-tuned dataset? More details about fine-tuning datasets should be included. Your fine-tuning significantly improves the accuracy. Is it because some structures are seen during training? Could you show some examples that, without fine-tuning, it fails, but after fine-tuning, it succeeds?
* Based on the definition of an Eulerian circuit that only visits edge once, but the example in appendix A.7 (Fig.7), the first generated sequence visit VOUT-PMI-G twice. Could you explain on why it is not consistent with your definition?

---

> ### Author Response · Authors · 2024-11-22
> **Author Response 1/5**
>
> We appreciate the reviewer's multiple valuable comments. Since most questions raised by the reviewer are on performance optimization and device sizing of analog circuits, we would like to make an overall clarification here on AnalogGenie before stepping into the detailed responses about addressing your specific questions. It is important to note that while performance optimization and device sizing based on well-established circuit topologies are important problems in analog circuit design automation, **our AnalogGenie focuses on generating novel topologies with the potential to outperform conventional ones** and this topology generation is the prerequisite for device sizing, presenting **the most significant challenge in analog design automation.** Yet, AnalogGenie is a performance-driven topology generation framework with **a typical optimization algorithm as its sizing algorithm (i.e., a generic algorithm for its model-free simplicity)** used in the fine-tuning step. More advanced optimization algorithms [1-10] can be integrated into fine-tuning, which will be explored in the future.
> > **Q1:** The main weakness of this paper is that it only considers generating functional circuits, which limits its effectiveness. In real-world tasks, analog designers need to optimize the topology based on the design targets, e.g., sizing. Does this work consider performance optimization during the generation process, or is it only considered functional? Have you performed sizing?
>
> **Yes, AnalogGenie considers performance optimization during the generation process rather than being limited to generating functional circuits. This is realized in the fine-tuning process.** Below, we provide additional details about its ability to target high-performance circuits effectively.
>
> The pre-trained AnalogGenie model does not exhibit an inherent preference for specific circuit types or performance metrics. Instead, it generates a diverse range of analog circuit topologies randomly by learning our large dataset, a similar functionality akin to pre-trained GPT. However, in practical applications, we aim to guide AnalogGenie to target specific types of analog circuits optimized for particular performance metrics (e.g., Figure of Merit, FoM). To achieve this, we utilize **reinforcement learning with human feedback (RLHF)** to fine-tune the pre-trained model, aligning its outputs with human-defined preferences for circuit type and performance. **The finetuning begins by training a reward model that evaluates newly generated topologies by assigning a score in terms of correctness, novelty, and performance. To train this model, we perform sizing using GA and SPICE simulation to label the dataset based on their type and performance.** Once the reward model is trained, we use **proximal policy optimization (PPO)** [18] to fine-tune AnalogGenie. In each epoch, the pre-trained model generates a batch of **new topologies**, and the reward model evaluates and scores them. PPO is then applied to optimize the model, maximizing the expected accumulated reward scores. **After finetuning, AnalogGenie is able to discover circuit topologies that are highly likely to achieve high performance once sized. Then, we perform sizing using GA and perform SPICE simulation to determine its exact FoM.**

---

> ### Author Response · Authors · 2024-11-22
> **Author Response 2/5**
>
> > **Q2:** To ensure the practicality of the designed circuit in real-world circuit design scenarios, the process technology of circuit fabrication should be taken into consideration. How do you think this current work can be extended to incorporate advanced process technologies and further optimize the design for improved performance and efficiency?
>
> **AnalogGenie has already taken technology in the reward model training dataset with BSIM4 45 nm. The performance of AnalogGenie after fine-tuning primarily depends on the reward model, which is trained on performance-labeled topologies that are already connected to a technology.** To create these performance labels, we utilize a GA to optimize circuit parameters, such as sizing and bias, and to determine the Pareto front of circuit performance represented as the Figure of Merit (FoM). Note that other optimization algorithms, such as reinforcement learning and Bayesian optimization, also work well. These optimizations are performed using BSIM4 45 nm technology models and NGSPICE as the simulation engine. This ensures that the labels account for realistic conditions. Through fine-tuning with the reward model trained by this labeled dataset, AnalogGenie has demonstrated the ability to generate high-performance circuits that operate effectively under real-world operation scenarios.
>
> AnalogGenies' pre-trained model is technology-independent, while the fine-tuned model is technology-dependent since it relies on labeled datasets that directly connect to technology. To extend AnalogGenie to advanced technology nodes without relabeling too many topologies, one can think about adopting transfer learning as an option. By collecting a limited number of labeled samples in the target technology node, we can fine-tune the reward model to adapt to the new technology. Transfer learning has demonstrated excellent sample efficiency in previous works [19-21] and offers a promising path to effectively apply AnalogGenie across a variety of advanced technology nodes.
>
> > **Q3:** Based on my understanding, the paper didn't propose methods to optimize circuits's FoM. Therefore, the paper shouldn't compare the FoM, as the benefits may come from your large dataset, and the model may reuse some circuits. This is unfair as previous methods cannot access the large dataset you have collected.
>
> **We would like to clarify that our FoM comparison is reasonable and fair.** To evaluate circuit performance, we begin by generating 100 circuit topologies for each method. We then apply a rule-based filtering process to exclude any circuits that are either invalid or duplicates of seen circuits in our dataset. Specifically, to determine if a topology is seen, we convert the generated sequence back into a graph representation and use a graph isomorphism algorithm to check for matches against circuits in our dataset. This ensures that all reported FoM values are derived from genuinely unseen topologies. After filtering, we use a GA to optimize the sizing of each circuit and report the maximum FoM obtained for comparison. **This approach ensures that all evaluated circuits are unseen and have been optimized for performance.**
>
> **When we reproduce our baseline in our experiment, all of them use the same large dataset we proposed. Yet, they are still unable to achieve the same level of performance as AnalogGenie did, due to the limitations of their methods.** Specifically, AnalogCoder is a training-free method based on existing GPT models and uses prompt engineering to generate analog circuits. We exactly follow the original work to produce the baseline result without any additional training. The training-free nature limits its generation capability. CktGNN and LaMAGIC adopt a top-down generation strategy that relies on a predefined representation with a fixed number of devices. **While we try to use all of our datasets, their representation significantly restricts the number of topologies that can be used for training, as most circuits in our dataset contain over 20 devices, exceeding the maximum limit these methods can handle.** The inherent limitations of prior methods restrict the variety of topologies they can learn. **Consequently, their poor performance relative to AnalogGenie is due to its method's limitation instead of an unfair experiment setup.**

---

> ### Author Response · Authors · 2024-11-22
> **Author Response 3/5**
>
> > **Q4:** The paper tries to tackle a simple task with topology generation, but how can the more critical analog circuit structure selection and sizing problem be solved?
>
> It is well known that topology generation is the prerequisite for device sizing, presenting the most significant challenge in analog design automation.
> This has been shown in many relevant research. More importantly, there have been extensive studies on device sizing [1-10] and analog circuit structure selection [11-15]. Yet, the research on topology discovery is limited [16-17]. Nonetheless, **AnalogGenie has already tackled circuit structure selection and sizing in its framework.** During generation, AnalogGenie selects the next device pin, which is the smallest structure in a circuit topology. It can also select the pin from a large analog structure like a phase-frequency detector (PFD) or from a digital gate such as a transmission gate as defined in **Appendix.2**. Higher-level structure selection can be further explored in the future and is beyond the scope of this work.
>
> For sizing, as we discussed in the beginning overview, AnalogGenie uses a genetic algorithm to perform circuit parameter optimization. But it's not the main focus of this work, and we will also explore more advanced optimization algorithms [1-10] in the future.
>
> > **Q5:** For fine-tuning, how do you ensure the validation circuit has no overlapped knowledge in the fine-tuned dataset? More details about fine-tuning datasets should be included. Your fine-tuning significantly improves the accuracy. Is it because some structures are seen during training? Could you show some examples that, without fine-tuning, it fails, but after fine-tuning, it succeeds?
>
> **As discussed in Q1, we do not use an external dataset for fine-tuning the pre-trained model; thus, there is no concern about overlapping knowledge. We adopt an iterative self-learning approach to finetune the pretrain model.** The pre-trained model generates new topologies in each iteration, which are then scored by the reward model. This process continuously **expands the training data**, allowing AnalogGenie to align its topology generation with high-performance circuit designs. Through this iterative fine-tuning, the model achieves improved correctness, novelty, and performance compared to the pre-trained model.
>
> To provide further clarification, we have added **Appendix 5** to showcase examples of failed results generated by the pre-trained model. In contrast, **Appendix 4** contains successful examples generated by the fine-tuned model. The pre-trained model’s failures often result in chaotic topologies that combine structural elements from multiple circuit types into a single circuit, leading to invalid designs. On the other hand, the fine-tuned model produces topologies that are more structured and aligned with human design preferences for specific circuit types. These fine-tuned topologies not only exhibit better design coherence but also achieve excellent performance after sizing and SPICE simulation.

---

> ### Author Response · Authors · 2024-11-22
> **Author Response 4/5**
>
> > **Q6:** Based on the definition of an Eulerian circuit that only visits edge once, but the example in appendix A.7 (Fig.7), the first generated sequence visit VOUT-PMI-G twice. Could you explain on why it is not consistent with your definition?
>
> **Our example in Appendix A.7 (Fig.7) aligns with our Eulerian circuit definition.** In **Definition 3.2.1**, an Eulerian circuit is defined as a trail in a graph that visits every edge exactly once and starts and ends at the same node. Specifically, for a directed graph, the Eulerian circuit visits every directed edge exactly once, while for an undirected graph, it visits every undirected edge exactly once. As detailed in **Section 3.2**, our process begins by representing each topology as a finite, connected, undirected graph. We then convert this graph into a directed graph by replacing each undirected edge $\{u, v\} \in E$ with two directed arcs $(u, v)$ and $(v, u)$ in $A$. Using depth-first search (DFS), we construct the Eulerian circuit for this directed graph, ensuring that each directed edge is traversed exactly once. Regarding the "duplicated edges" observed in **Appendix A.7**, such as `'PM1_G' -> 'VOUT1'` and `'VOUT1' -> 'PM1_G'`, these represent distinct directed edges in the graph and are consistent with our definition, as each directed edge is uniquely traversed. We would also like to emphasize that our claim about the space advantage of Eulerian circuits over adjacency matrices remains valid. Adjacency matrices also treat undirected graphs as directed graphs for storage purposes, requiring both `'PM1_G' -> 'VOUT1'` and `'VOUT1' -> 'PM1_G'` to be stored as separate elements. Thus, using an Eulerian circuit provides a more compact and efficient representation while retaining all necessary connectivity information. We hope this clarification addresses your concerns and confirms the consistency of our approach with the stated methodology.
>
> Although our Eulerian circuit definition is consistent with our example, we identified an unintended typo upon reviewing the example in **Appendix A.7**. The provided Eulerian circuit example was based on an outdated graph representation from our previous study. Specifically, it omits several edges that are present in the most recent representation of the graph. The missing undirected edges include `'PM1_G' <-> 'PM1_D'`, `'PM1_G' <-> 'NM1_D'`, `'NM1_B' <-> 'NM1_S'`, `'PM1_B' <-> 'PM1_S'`, and `'PM1_D' <-> 'NM1_D'`. These edges are crucial for explicitly modeling all connections between circuit pins. We have corrected this error by updating **Figure 7** and revising the accompanying text in **Appendix A.7** to reflect the current graph representation. **We sincerely appreciate your question, as it allowed us to identify and address this oversight.**
>
>
>
> We have made every effort to address Reviewer sJmh's concerns and hope our response meets your expectations, potentially leading to a higher rating.

---

> ### Author Response · Authors · 2024-11-22
> **Author Response 5/5**
>
> [1] Budak, Ahmet F., et al. "Dnn-opt: An rl inspired optimization for analog circuit sizing using deep neural networks." *2021 58th ACM/IEEE Design Automation Conference (DAC)*. IEEE, 2021.
>
> [2] Wang, Hanrui, et al. "GCN-RL circuit designer: Transferable transistor sizing with graph neural networks and reinforcement learning." *2020 57th ACM/IEEE Design Automation Conference (DAC)*. IEEE, 2020.
>
> [3] Lyu, Wenlong, et al. "An efficient bayesian optimization approach for automated optimization of analog circuits." *IEEE Transactions on Circuits and Systems I: Regular Papers* 65.6 (2017): 1954-1967.
>
> [4] Cao, Weidong, et al. "Domain knowledge-infused deep learning for automated analog/radio-frequency circuit parameter optimization." *Proceedings of the 59th ACM/IEEE Design Automation Conference*. 2022.
>
> [5] Shi, Wei, et al. "Robustanalog: Fast variation-aware analog circuit design via multi-task rl." *Proceedings of the 2022 ACM/IEEE Workshop on Machine Learning for CAD*. 2022.
>
> [6] Kong, Zichen, et al. "PVTSizing: A TuRBO-RL-Based Batch-Sampling Optimization Framework for PVT-Robust Analog Circuit Synthesis." *Proceedings of the 61st ACM/IEEE Design Automation Conference*. 2024.
>
> [7] Gu, Tianchen, et al. "Bbgp-sdfo: Batch bayesian and gaussian process enhanced subspace derivative free optimization for high-dimensional analog circuit synthesis." *IEEE Transactions on Computer-Aided Design of Integrated Circuits and Systems* (2023).
>
> [8] Zhao, Aidong, et al. "cvts: A constrained voronoi tree search method for high dimensional analog circuit synthesis." *2023 60th ACM/IEEE Design Automation Conference (DAC)*. IEEE, 2023.
>
> [9] Zhang, Jinxin, et al. "Automated Design of Complex Analog Circuits with Multiagent based Reinforcement Learning." *2023 60th ACM/IEEE Design Automation Conference (DAC)*. IEEE, 2023.
>
> [10] Gao, Jian, Weidong Cao, and Xuan Zhang. "RoSE: Robust Analog Circuit Parameter Optimization with Sampling-Efficient Reinforcement Learning." *2023 60th ACM/IEEE Design Automation Conference (DAC)*. IEEE, 2023.
>
> [11] Zhao, Zhenxin, and Lihong Zhang. "Analog integrated circuit topology synthesis with deep reinforcement learning." *IEEE Transactions on Computer-Aided Design of Integrated Circuits and Systems* 41.12 (2022): 5138-5151.
>
> [12] Abel, Inga, and Helmut Graeb. "FUBOCO: Structure synthesis of basic op-amps by FUnctional BlOck composition." *ACM Transactions on Design Automation of Electronic Systems (TODAES)* 27.6 (2022): 1-27.
>
> [13] Zhao, Zhenxin, and Lihong Zhang. "An automated topology synthesis framework for analog integrated circuits." *IEEE Transactions on Computer-Aided Design of Integrated Circuits and Systems* 39.12 (2020): 4325-4337.
>
> [14] Lai, Yao, et al. "AnalogCoder: Analog Circuit Design via Training-Free Code Generation." *arXiv preprint arXiv:2405.14918* (2024).
>
> [15] Chang, Chen-Chia, et al. "Lamagic: Language-model-based topology generation for analog integrated circuits." *arXiv preprint arXiv:2407.18269* (2024).
>
> [16] Dong, Zehao, et al. "CktGNN: Circuit graph neural network for electronic design automation." *arXiv preprint arXiv:2308.16406* (2023).
>
> [17] Karahan, Emir Ali, Zheng Liu, and Kaushik Sengupta. "Deep-learning-based inverse-designed millimeter-wave passives and power amplifiers." *IEEE Journal of Solid-State Circuits* 58.11 (2023): 3074-3088.
>
> [18] Ouyang, Long, et al. "Training language models to follow instructions with human feedback." *Advances in neural information processing systems* 35 (2022): 27730-27744.
>
> [19] Zhang, Qiaochu, et al. "CEPA: CNN-based early performance assertion scheme for analog and mixed-signal circuit simulation." *Proceedings of the 39th International Conference on Computer-Aided Design*. 2020.
>
> [20] Liu, Juzheng, et al. "Transfer learning with Bayesian optimization-aided sampling for efficient AMS circuit modeling." *Proceedings of the 39th International Conference on Computer-Aided Design*. 2020.
>
> [21] Shahane, Aditya, et al. "Graph of circuits with GNN for exploring the optimal design space." *Advances in Neural Information Processing Systems* 36 (2024).

---

> ### Comment · Reviewer_sJmh · 2024-11-25
> **Thank you so much for your responce**
>
> I have read all the responses, and I have no further questions or concerns. Overall, it is a good paper and may truly advance the analog design automation field. I will raise the point. Good work!

---

> > ### Author Response · Authors · 2024-11-25
> >
> > Thanks for recognizing our work and raising the score. Feel free to let us know if you have any questions.

---

### Author Response · Authors · 2024-11-22
**General Author Response**

We sincerely thank all reviewers for their time, effort, and constructive feedback on our paper. We appreciate the reviewers' unanimously positive reception of our work in the two major contributions:

(1) novel sequential graph representation for representing analog circuit

- Quote from Reviewer sJmh "The idea of transferring the analog circuit into a sequence-based graph representing is a novel and interesting idea, enabling the use of a generative model for scalable circuit generation."
- Quote from Reviewer U6NJ "The authors avoid using adjacency matrices to represent circuit topologies. Instead, they employ an Eulerian graph representation, which effectively reduces the model’s memory requirements."
- Quote from Reviewer KSXW "developing a scalable sequence-based graph representation."
- Quote from Reviewer nrPJ " The authors treat undirected edges as two directed arcs in opposite directions, ensuring that the graph contains at least one Eulerian circuit starting from any vertex. Using this approach, they can sequentialize the graph of an analog circuit into an Eulerian circuit—a trail that visits each edge exactly once and both starts and ends at the "VSS" node."

and (2) creating and curating a large-scale analog circuit dataset.

- Quote from Reviewer sJmh "The collection of large-scale analog circuits. It would be great if the authors can release the dataset."
- Quote from Reviewer U6NJ "The authors present a new open-source dataset that comprises LDOs, bandgap references, comparators, PLLs, LNAs, PAs, mixers, and VCOs. Unlike previous datasets, this dataset includes performance metrics for each circuit. To ensure accuracy, the researchers manually implemented each circuit in an industry-standard design tool and conducted simulations, which is a significant manual effort.."
- Quote from Reviewer KSXW "AnalogGenie takes a novel approach by creating a large dataset of over 3,000 analog circuit topologies."
- Quote from Reviewer nrPJ " Additionally, this work provides a comprehensive, high-quality, large-scale dataset, poised to greatly advance analog design automation."

For the questions and concerns raised by the reviewers, we have spared no effort to address them. We hope our explanations and additional clarifications have resolved any concerns. Feel free to let us know if you have follow-up questions. We are happy to address them. Additionally, we have revised the manuscript accordingly, with all modifications clearly highlighted in blue for transparency.

---

### Meta-Review · Area_Chair_jkHR · 2024-12-17

**Metareview:**

This paper presents an automatic discovery technique for analog circuit topologies, which the authors call AnalogGenie. To develop this system, the authors collected a large and complete analog circuit dataset, developed a scalable method for expressing analog circuits as a sequence-based graph, then trained a generative model. The authors claim that AnaolgGenie is capable of automatically generating better large-scale, valid, unseen, and high-performance analog circuit topologies compared to existing systems.

The strengths of this paper are that it assembles an important dataset and uses that data, coupled with several novel innovations, to presents a real step-forward in analog circuit generation. The weaknesses are that the analysis of the performance of the generated circuits is not as fleshed out as it could be, and the discussion of limitations was not as robust as it could have been either. But, on balance, this is a solid paper of real interest to engineers concerned with integrated ciruits, and so a decision to accept (poster) was reached.

**Additional Comments On Reviewer Discussion:**

The discussion was courteous and productive, with the authors addressing many of the reviewers concerns. As such, the final scores were an average of 7.5, making this a clear accept. The reviewers did not voice any desire in discussion for this to be a spotlight, though, and given that it is relatively niche as a topic, a poster seemed natural.

---

### Decision · Program_Chairs · 2025-01-22

Accept (Spotlight)